



# The influence of long-term changes in canopy structure on rainfall interception loss: a case study in Speulderbos, the Netherlands.

César Cisneros Vaca, Christiaan van der Tol, Chandra Prasad Ghimire

Faculty of Geo-information and Earth Observation (ITC), University of Twente, Enschede, P.O. Box 217, 7500 AE, the Netherlands

*Correspondence to*: César Cisneros Vaca (c.r.cisnerosvaca@utwente.nl)

**Abstract.** The evaporation of intercepted water by forests is a significant contributor to both the water and energy budget of the Earth. In many studies, a discrepancy in the water and energy budget is found: the energy that is needed for evaporation is

larger than the available energy supplied by net radiation. In this study, we analyse the water and energy budget of a mature Douglas-fir stand in the Netherlands, for the two growing seasons of 2015 and 2016.  Based on the wet-canopy water balance equation for these two growing seasons, derived interception losses were estimated to be 37% and 39% of gross rainfall, respectively.

We further scrutinized eddy covariance energy balance data from these two consecutive growing seasons and found the average

evaporation rate during wet canopy conditions was 0.20 mm h$^{-1}$. The source of energy for this wet-canopy evaporation was net radiation (35 %), a negative sensible heat flux (45 %) and a negative energy storage change (15 %). This confirms that the energy for wet-canopy evaporation is extracted from the biomass as well as the atmosphere.

Moreover, the measured interception loss at the forest was similar to that measured at the same site years before ($I$ = 38 %), when the forest was younger (29 years old, vs 55 years old in 2015). At that time, the forest was denser and had a higher

canopy storage capacity (2.4 mm then vs 1.90 mm in 2015), but the aerodynamic conductance was lower (0.065 m s$^{-1}$ then vs 0.105 m s$^{-1}$ in 2015), and therefore past evaporation rates were lower than evaporation rates found in the present study (0.077 mm h$^{-1}$ vs 0.20 mm h$^{-1}$ in 2015). Our findings emphasize the importance of quantifying downward sensible heat flux and heat release from canopy biomass in tall forest in order to improve the quantification of evaporative fluxes in wet canopies.

## 1 Introduction

Rainfall interception is the portion of precipitation temporarily captured by vegetation before evaporating back into the atmosphere. It is by definition unavailable for soil infiltration or run-off. Evaporation from intercepted rainfall is an important component of the water balance, and in coniferous forests it can represent around 25-45 % of gross rainfall ($P_G$) (Rutter et al., 1975; Gash et al., 1980; Carlyle-Moses and Gash, 2011). Starting from the early twentieth century (Horton, 1919; benchmark papers in Gash and Shuttleworth, 2007) much research has been done to quantify the magnitude of rainfall interception over



different forest ecosystems (cf. Carlyle-Moses and Gash, 2011; Muzylo et al., 2009; Miralles et al., 2010; Llorens and Domingo, 2007). Despite decades of research, the physical processes and atmospheric conditions that allow a large fraction of rainfall to return to the atmosphere are still poorly understood (van Dijk et al., 2015). A key issue is that the water budget often suggests a higher evaporation rate than the available energy permits. In addition, little research has focused on the long-term

evolution of rainfall interception loss with forest growth and development, and on the implications for the forest water balance. Improving knowledge of the evolution of rainfall interception with forest growth, in particular the canopy storage and wet canopy evaporation rate, will provide insight into the role of forests regarding moisture recycling, and will assist when developing both forest management strategies against threats to forest water resources, and remote sensing techniques to monitor rainfall interception based on forest structure (i.e. Miralles et al., 2010; Cui et al., 2015; Hassan et al., 2017).

In this study, we revisited a site with a Douglas-fir plantation in the centre of the Netherlands, the 'Speulderbos', for which historical measurements of rainfall interception are available. During the 1990's research focused on the impact of air pollution on forest growth ('Aciforn project') at this site (Evers et al., 1991a). Several hydrological studies at that time found high interception losses (~40 %) from the forest (Tiktak and Bouten, 1994; Klaassen et al., 1998; Bouten et al., 1996), contributing to the debate about forest plantations in the centre of the Netherlands affecting the ground water recharge (Moors, 2012).

Detailed measurements and modelling studies at the site included canopy growth and architecture (Evers et al., 1991b), soil water dynamics (Tiktak and Bouten, 1994), modelling evaporation and transpiration (Bosveld and Bouten, 2001), spatial patterns of throughfall (Bouten et al., 1992), and measuring and modelling canopy water storage (Bouten et al., 1991; Bouten et al., 1996). It was found that the high interception losses in Speulderbos at that time were due to a high leaf area index, resulting in high canopy storage capacity (2.5 mm) as measured by microwave transmission (Bouten et al., 1991). Evaporation

during rainfall contributed minimally to the overall interception loss (Klaassen et al., 1998).

The historical data collected in Speulderbos offer a good opportunity to evaluate the effects of long-term changes in canopy structure on the rainfall interception process and their implications for other phases of the water balance (i.e. soil infiltration, plant water uptake, ground water recharge, stream discharge). Changes in forest structure can be resultant from different factors as tree phenology, management practices (Bormann et al., 2015), changes in species composition (Thom et al., 2017), and

stand development (Franklin et al., 2002, Freund et al., 2015). Although several studies have analysed the effects of short-term changes (i.e. seasonal) in forest canopy structure on rainfall interception (Dolman, 1987; Price and Carlyle-Moses, 2003; Deguchi et al., 2006; Herbst et al., 2008; Muzylo et al., 2012), much fewer studies have considered the effects of forest growth over longer time scales (i.e. decades, centuries) in forest canopy structure on rainfall interception. In some cases comparisons have been limited to monitoring the water balance components of two (or more) stands of different ages concurrently (i.e.

Pypker et al., 2005; Keim et al., 2005).

Due to the long-term maintenance of the research infrastructure at the site, a unique opportunity arose to evaluate the changes in eco-hydrological processes over several decades. We were particularly interested in the water and energy budget of the forest. For this, two variables are of key importance: the water storage capacity of the forest, and the evaporation rate of the wet canopy, which is energy limited. The evaporation rate in turn, depends on the radiation budget, the aerodynamic roughness



and the heat storage capacity. We expected that thinning of the forest stand in combination with a substantial increase in forest height must have affected the water and energy storage capacity, as well as the forest roughness.

For this study, we combined data collected using an eddy covariance flux tower with precipitation, stemflow and throughfall data to obtain an estimate of the interception loss from a mature Douglas-fir plantation (ca. 55 years old).. The objectives of the present study were to:

i)      assess two indirect methods for estimating canopy storage capacity,

ii)     evaluate the sources of the latent heat flux involved in the evaporation of intercepted rainfall,

iii)    analyse the interception loss process for two consecutive growing seasons (in 2015 and 2016) for a mature Douglas-fir forest,

iv)     use a physically based interception model to explore the different processes contributing to rainfall interception loss.

## 2 Materials and Methods

### 2.1 Research Site

The study was conducted within a 2.5 ha evergreen Douglas-fir (*Pseudstuga menziesii*) stand located in the forested area of 'Speulderbos' (52° 15' 04"N, 05° 41' 25"E) at an elevation of 50 m.a.s.l., near the settlement of Garderen, the Netherlands (Fig. 1). The site is equipped with a 47 m scaffolding tower, which supports measurement of a range of micrometeorological data. The plot is surrounded by several stands of other species such as beech, oak and hemlock. The climate is classified as temperate-humid. Based on 'de Bilt' weather station data, located at 38 km SW of the plot, the average (±SD) annual precipitation for the period 2000-2015 was 864 (±92) mm. In general, July is the wettest month with about 12 % of the annual rainfall and April the driest month with 4 % of the annual rainfall. The mean annual temperature is 10.6 °C (±0.6) with January being the coldest month (3.7 ±2 °C) and July the warmest month (18.2 ±1.6 °C) (KNMI, 2015). The type of soil in the study area is Typic Dystochrepts on thick heterogeneous sandy loam and loamy sand textured ice-pushed river sediments (Tiktak and Bouten, 1988).

Active reforestation in the area, previously sand dunes, started at the end of the 19[th] century. The current stand was planted with two-year-old seedlings in 1962. For the study period canopy height was about 34 m, whereas stem density and mean DBH were 571 trees ha[-1] and 34.8 (±8.9) cm, respectively. The leaf area index of the plot (LAI, using a LI-COR LAI 2000 Plant Canopy Analyser) was 4.5 (±0.38) (Fig. 1b). No other tree species were recorded in the plot and understory was largely absent (Fig. 1c).



### 2.2 Field measurements

#### 2.2.1 Rainfall

Gross rainfall ($P_G$, mm) was measured in a nearby well-exposed clearing (ca. 250 m from the centre of the plot) using 2 tipping bucket rain gauges (Rain Collector II, Davis Instruments, USA) with a resolution of 0.2 mm per tip. The orifice of the rain gauge was positioned at 1.5 m above the ground to avoid ground-splash effects. The automatically recorded data were stored by a HOBO event logger at 1 min intervals (Onset Computer Corporation, USA).

Gross rainfall was also collected at the top of the 47 m scaffolding tower operated by University of Twente (ITC-UT) (at ca. 200 m distance from the clearing), using a tipping bucket rain gauge (Onset HOBO-RG3, resolution 0.2 mm). The data collected at the top of the tower were only used to fill gaps in the data from the clearing (from 23 July 2015 to 12 August 2015, as well as 24 May 2016 to 09 June 2016) using a linear regression equation linking 10 min rainfall totals of the two locations ($R^2 = 0.93$, $n = 500$).

#### 2.2.2 Throughfall

Throughfall ($Tf$, mm) was measured by an automated gutter system and validated by an arrangement of manual (roving sampling) funnel-type collectors. The automated gutter system consisted of four stainless steel gutters (200 cm x 30 cm each), randomly located in the plot and connected by pairs to two tipping buckets (V2A UP Umweltanalytische Produkte GmbH). As no apparent alignment of the trees was observed in the planted stand, no specific orientation of the gutters was considered. The gutters were mounted on a wooden frame, about 60 cm from the forest floor and at an inclination of 10 % to facilitate drainage to the tipping buckets. Combining two gutters and correcting for the inclination provided a total catch area of 1.2 m² yielding 0.084 mm per tip. The tipping buckets were connected to a data logger (CR23X, Campbell Scientific Ltd.) and tip pulses were recorded at a 1 min resolution. The gutters and the tubes were cleaned every 7 to 15 days to avoid clogging due to falling litter. In addition, $Tf$ was measured using funnel-type collectors. The manual array of collectors was operated from 17 February 2015 to 02 November 2015. A stratified random sampling approach was used to ensure an even spread of sampling locations. We defined a plot size of 32 x 64 m, which was divided into 32 square sub-plots of 8 m x 8 m each; each sub-plot was marked in its centre. Collectors (32 in total) were placed at some distance from each marked point, by generating random values for an azimuth angle and distance from the grid point. The azimuth angles ranged from 0 to 360 degrees and the distances from 0 to 4 m. In case the randomly selected position coincided with the position of a stem, the azimuth angle was maintained while the distance was adjusted until the collector was located next to the tree base and the adjusted distance was recorded. The funnel-type collectors consisted of a 2 L collector and a funnel (165 cm² orifice area). The orifices of the gauges were positioned 50 cm above the forest floor to avoid splash-in from the ground. The funnel-type collectors were read (and relocated) ~ bi-weekly (i.e. roving sampling; Ritter and Regalado, 2014). Measured $Tf$ volumes were converted to equivalent depth (in mm) by dividing the volume of water in each gauge by the orifice area.





Using the manual array, $Tf$ was measured 15 times (cf. Table A1). For the first ten measurement periods we applied a roving sampling method by randomly relocating the position of the funnel-type collectors after each $Tf$ measurement (resulting in 320 different positions of the funnel type collectors). However, the coefficient of variation (CV) in the ten initial measurement periods was low (~15 %) and therefore we did not use the roving technique for the remaining five measurement periods (i.e., the gauges were not relocated after measurements were taken).

### 2.2.3 Stemflow

Following a stratified sampling approach, stemflow ($Sf$, mm) was measured on 4 trees with differing diameter at breast height (DBH), representative for the whole stand. The four diameter size classes were: < 30 cm, 31-40 cm, 41-50 cm and >50 cm. Each set-up consisted of a halved plastic tube wrapped around the tree stem in a spiral fashion, starting at a height of ca. 80 cm, the lower end of the tube was connected to a closed tipping bucket (Onset HOBO® S-RGA Rain Gauge, resolution 0.254 mm). Silicon sealant was applied between the stem and the plastic tube to seal the gaps (and hence avoid stemflow loss). Stemflow proved to be only a minor component of the wet canopy water balance. As the sampled trees covered the whole range of diameter classes within the plot, total stemflow in the plot was calculated by multiplying the stemflow volumes by the number of trees for each diameter class (c.f. Levia and Germer, 2015; Eq. (2)). Stemflow measurements were carried out over 113 days from 27 July to 11 November, 2016. During this period, a total of 240 mm of rain was received at the plot.

### 2.2.4 Net radiation and soil heat flux

Net radiation ($R_n$) was measured by a four-component net radiometer (model CNR1, Kipp and Zonen) mounted at 35 m above ground (Table 1), and averages were stored at 10 minute intervals, except during three periods, totalling 120 days, encountering instrument failure (from 24 April to 23 June, 2016, from 17 July to 04 August, 2016, and from 15 September to 14 October, 2016).

Soil heat flux ($G$) was estimated by combining measurements from two heat flux plates (HFP01, Hukseflux) both installed at 8 cm depth, and temperature profile measurements at five different depths (1, 3, 8, 20, and 50 cm). Estimations of $G$ at ground level were carried out following the harmonics method (Verhoef et al., 1996) with the derivatives of the Fourier series calculated analytically according to van der Tol (2012).

### 2.2.5 Turbulent heat fluxes

Sensible ($H$) and latent ($\lambda E$) heat flux were estimated by the eddy-covariance technique with a sonic anemometer (CSAT3, Campbell Sci. Inc.) and an open path gas analyser (LI7500, Li-Cor Biosciences) mounted at 47 m (Table 1).

Thirty minute turbulent fluxes were processed with the software EddyUH ver. 1.7 (University of Helsinki, https://www.atm.helsinki.fi/Eddy_Covariance/EddyUHsoftware.php). As initial estimates for EddyUH, a displacement height ($d$) of 0.7 of canopy height ($h$) (Stull, 2012), and a roughness length ($z_0$) of 0.06 $h$ (Weligepolage et al., 2012) were used.



Furthermore, the following corrections were performed: de-spiking, 2D coordinate rotation, cross-wind correction to sonic temperature according to Liu et al. (2001), high frequency spectral corrections according to Moncrieff et al. (1997) and Aubinet et al. (2000), low frequency spectral corrections according to Rannik and Vesala (1999), correction for humidity effect on sonic heat flux according to Schotanus et al. (1983), and WPL correction according to Webb et al. (1980). We disregarded

5 turbulence data with an overall quality flag above 2 in accordance with the Foken et al. (2005) quality flag system.

### 2.2.6 Energy storage

Energy storage ($Q$), composed of energy storage in the canopy air ($Q_{air}$) and in the biomass ($Q_{bio}$), was estimated based on measurements of a vertical profile of air temperature and humidity. Further details on the sensors and their location are shown in Table 1. Energy storage changes in the air $Q_{air}$ result from the change in the temperature in the air $Q_T$ plus the component resulting from the specific humidity $Q_q$ (cf. Michiles and Gielow, 2008).

$Q_{bio}$ was estimated as the sum of energy stored in the trunks ($Q_{tr}$), in the branches ($Q_{br}$), and in the needles ($Q_{nd}$). For Douglas-fir in the centre of the Netherlands, allometric equations on biomass and its vertical distribution derived from Bartelink (1996) were used (Table 2). Specific heat of biomass ($c_v$) was assumed to be equal for all components at 2400 J kg$^{-1}$ C$^{-1}$ (cf. Ringgaard et al., 2014). A moisture content of 44 % for Douglas-fir (cf. Nord-Larsen and Nielsen, 2015) was used to estimate the dry

15 matter content.

$Q_{air}$ was estimated by dividing the air column into four sections: Section 1 (0 to 10 m), Section 2 (10 to 20 m), Section 3 (20 to 28 m) and Section 4 (28 to 34 m). Each section was centred on the respective level of temperature and humidity (Table 1) considered representative for the entire section. The following equations suggested by McCaughey (1985), were used to estimate $Q_{air}$:

$$Q_{air} = Q_T + Q_q \tag{1}$$

$$Q_{air} = \rho\left(c_p \Delta\bar{T} + \lambda\Delta\bar{q}\right)\Delta z/\Delta t \tag{2}$$

where $\rho$ (kg m$^{-3}$) is density of air, $c_p$ (J kg$^{-1}$ K) is the specific heat of air, $\lambda$ (J kg$^{-1}$) is the latent heat of vaporization, $\Delta\bar{T}$ (K) and $\Delta\bar{q}$ (kg kg$^{-1}$) are the change in mean air temperature and specific humidity over time, respectively, $\Delta z$ (m) is the height

25 thickness of the considered layer, and $\Delta t$ (s) is the time interval.

$Q_{bio}$ was estimated by means of the following equation (Oliphant et al., 2004; McCaughey, 1985):

$$Q_{bio} = m_{bio} c_v \Delta T_{bio}/\Delta t \tag{3}$$





where $m_{bio}$ (kg m$^{-2}$) is the mass of biomass per unit of horizontal area, $c_v$ (J kg$^{-1}$) is a representative specific heat of the vegetation and $T_{bio}$ (K) is a representative biomass temperature. Some studies have used ambient air temperature as a surrogate for $T_{bio}$ (Oliphant et al., 2004; Thom et al., 1975; Michiles and Gielow, 2008). We used Eq. (3) with $T_{bio}$ equalling air temperature for intervals without rainfall, and wet bulb temperature (Stull, 2011) for intervals with rainfall ($P_G > 0.5$ mm) (cf. van Dijk et al., 2015).

### 2.2.7 Canopy wetness

Three leaf wetness sensors (LWSs) (Model 237, Campbell Sci. Inc.) were installed at 20, 24, and 26 m above ground, in the middle of the living crown. Ringgaard et al. (2014) used 4 LWSs located within the canopy, and demonstrated that the parametrization of the analytical model for interception loss by Gash et al. (1995) was improved. A LWS consists of a circuit board that emulates the leaf surface, with interlacing gold-plated fingers on it. The LWS estimates foliage wetness by determining the electrical resistance on the surface of the sensor. Sensors were placed over the needles in the middle of the branches and tilted about 60° to avoid rainwater puddling on the electrodes.

## 2.3 Methods

### 2.3.1 Modelling rainfall interception

The Gash analytical model (Gash, 1979) was used to simulate rainfall interception loss (*I*, mm). The Gash model assumes that rainfall occurs as a series of discrete events. The Gash model differentiates three phases in a rainfall event: i) the canopy wetting phase, ii) the canopy saturation phase, iii) the canopy drying phase. Table 3 summarizes the equations associated with the respective phases. The Gash model uses four canopy parameters: i) canopy storage capacity $S$ , which is defined as the amount of water left in a saturated canopy in the absence of evaporation, after rainfall and drainage has ceased, ii) the free throughfall coefficient $p$, which is the fraction of incident rainfall that reaches the forest floor without touching the forest canopy, iii) the coefficient $p_t$, which is the fraction of rain diverted to the trunks as $Sf$, iv) stem storage capacity $S_t$, which is the amount of water that can be stored on the stems. In addition to the canopy parameters, the Gash analytical model requires two climatic parameters: the mean evaporation rate $\bar{E}$ and the mean rainfall intensity $\bar{R}$.

We divided our data-set into two parts: data-set 1 included measurements from 19 June to 31 October, 2015, and data-set 2 included measurements from 19 June to 31 October, 2016. Data-set 1 was used for parameter estimation (model calibration), and data-set 2 was used for validation.

We used two different parametrizations of the Gash model. In the first parametrization (Run 1) the parameters $S$, $p$ and $\bar{E}/\bar{R}$ were derived from the water balance (rainfall, throughfall and stemflow data) by using the mean method. In the mean method $S$, $p$ and $\bar{E}/\bar{R}$ were derived from linear regressions of measured $I$ versus measured $P_G$ from multiple events (Klaassen, 2001).





In the second parametrization (Run 2), the parameters $S$ and $p$ were derived using individual event analysis (IEA) (Link et al., 2004), while the parameter $\bar{E}/\bar{R}$ was calculated with $E$ derived from the energy balance residual and $R$ derived from the tipping bucket measurements. Because the distributions of $E$ and $R$ were skewed, the use of median rather than mean was preferred, (cf. Schellekens et al., 1999) For both parametrizations, values of $S_t$, and $p_t$ as derived with the Gash and Morton

(1978) method were used. The methods are discussed in detail in the following sections.

Moreover, the sensitivity of the modelled $I$ to the relevant parameters, namely $S$ and $\bar{E}$ (Gash, 1979; Loustau et al., 1992; Moors, 2012) was evaluated. We used the RMSE as the criterion to test the sensitivity for data-set 1 and data-set 2 separately. By comparing these different runs and the model sensitivity, we were able to evaluate the consistency of the water and energy balance estimates of evaporation as well as the storage capacity of the canopy.

**2.3.2 Derivation of canopy parameters**

Two methods were used to derive the canopy parameters: the multiple event analysis or mean method (Klaassen et al., 1998) and the individual event analysis (IEA) (Link et al., 2004). As the Gash (1979) model is event based, it is important to discriminate events first. Because the criteria used to discriminate events can have a major effect on the interception modelling, we evaluated two cases of event selection: Case A, considering an event as a period of rain exceeding 0.5 mm preceded by a

dry period of at least 3 h (cf. Klaassen et al., 1998), and Case B, considering an event as a period of rain exceeding 0.5 mm, with the preceding dryness validated by three LWSs (all indicating fully dry) within the living crown. In addition, for Case B saturated events were considered only those events with $P_G \geq 5$ mm.

The mean method is a multi-event analysis where $S$ and $p$ are estimated by linear regression of interception loss ($I = P_G - Tf - Sf$) versus $P_G$. To estimate $p$ the regression is in the form of $I = aP_G$ with $a = 1 - p - p_t$ and only events with a total

amount of rainfall unable to saturate the canopy ($P_G < P_G{}'$) are used. The parameters $S$ and $\bar{E}/\bar{R}$ are derived from events large enough to saturate the canopy ($P_G \geq P_G{}'$) with the linear regression in the form of $I = b_1 P_G + b_2$, where $b_1 = \bar{E}/\bar{R}$ and $b_2 = (1 - p - p_t)P_G{}'$. An iterative procedure is employed where the initial value of $P_G{}'$ is visually defined from an $I$ versus $P_G$ graph. After fitting both equations, $P_G{}'$ is re-calculated as $P_G{}' = b_2/(a + b_1)$ and the process is repeated until $P_G{}'$ converges (cf. Klaassen et al., 1998; Holwerda et al., 2012).

In contrast, the IEA consists of an analysis of the behaviour of water fluxes during individual events. It is based on the equations proposed in the Gash (1979) analytical model. When rainfall starts ($P_G < P_G{}'$) throughfall increases approximately linearly with $P_G$ ($Tf = pP_G$) until saturation is reached $P_G \geq P_G{}'$. When rainfall saturates the canopy storage capacity ($P_G = P_G{}'$) an inflection point in the $Tf$ vs. $P_G$ plot occurs and water starts to drain from the canopy. Canopy parameters $p$, $P_G{}'$ and $S$ can be derived using an iterative regression procedure over the plot of cumulative $Tf$ versus cumulative $P_G$. The procedure was

applied to events selected in Case B (i.e. using LWS and saturation threshold $P_G \geq 5$ mm) and with data records at a 15 min time resolution. Events that did not have sufficient 15 min records before saturation was reached were discarded. The inflection point ($P_G{}'$) was calculated as the intersection of two linear regressions, for the wetting-up stage and after canopy saturation.




### 2.3.3 Wet-canopy evaporation

Three methods were used to estimate wet-canopy evaporation rate, based on: i) a water balance approach, ii) the energy balance residual and iii) the Penman-Monteith equation.

Firstly, using the water balance approach, the average wet-canopy evaporation rate ($\bar{E}$) was derived from the value of $\bar{E}/\bar{R}$,

obtained from the mean method and from the median rainfall rate, as measured by the tipping bucket rain gauge. The use of median rather than mean was preferred, because the distribution of rainfall intensity was skewed (cf. Schellekens et al., 1999). The thus derived value of $\bar{E}$ will henceforth be referred to as the 'water balance based' evaporation rate.

Secondly, wet-canopy evaporation rates were estimated from the energy balance residual. The quality of the energy balance data (eddy-covariance flux of sensible heat, net radiation, ground heat and storage terms) was verified by calculating the energy

balance closure and the energy balance ratio (EBR) for the dry and wet periods for which high quality data (Quality flag $\leq 2$ ) of $\lambda E$ were available. In addition, the performance of the sonic anemometer (CSAT3) during wet conditions was evaluated by plotting the standard deviation of the vertical wind speed ($\sigma w$) against friction velocity ($u^*$) (Gash et al., 1999; van der Tol et al., 2003; Holwerda et al., 2012) for the full study period from 19 June 2015 to 31 October 2016, but excluding the winter season from November 2015 to March 2016.

Wet-canopy evaporation rate was derived using the energy balance residual approach where $\lambda E$ (W m$^{-2}$) is estimated as the residual of the energy balance equation as:

$$\lambda E = R_n - H - G - Q - G_P \tag{4}$$

with $H$ derived from the eddy-covariance technique and $G_p$, the photosynthetic energy flux, estimated at -2 W m$^{-2}$ during the

night and at 6 W m$^{-2}$ during the day (Thom et al., 1975). Equation (4) provides a more complete dataset than $\lambda E$ based on the eddy covariance data only, due to the fact that the open path gas analyser is prone to providing low quality data (causing rejections during filtering) during wet periods. The evaporation estimated with Eq. (4) is hereinafter referred as $E_{EB-EC}$.

Finally, the last and most common method to estimate wet canopy evaporation, the Penman-Monteith equation (P-M), was used. P-M estimates latent heat flux ($\lambda E$, W m$^{-2}$) as:

$$\lambda E = \frac{\Delta A + \rho c_p (e_s - e) g_a}{\Delta + \gamma'} \tag{5}$$

with:

$$\gamma' = \gamma \left(1 + \frac{g_a}{g_s}\right) \tag{6}$$



where $\Delta$ (hPa K$^{-1}$) is the slope of the saturated water vapour pressure curve, $A$ (W m$^{-2}$) is the available energy, $\gamma'$ and $\gamma$ (hPa K$^{-1}$) are the adjusted and original psychrometric constant, respectively, $\rho$ (kg m$^{-3}$) is the density of air, $c_p$ (J kg$^{-1}$ K$^{-1}$) is the specific heat of air at constant pressure, $e_s$ (hPa) is the saturation vapour pressure at ambient temperature, $e$ (hPa) is the actual vapour pressure, $g_a$ (m s$^{-1}$) is the aerodynamic conductance, and $g_s$ (m s$^{-1}$) is the surface conductance.

During wet conditions surface conductance is assumed to be infinitely large (i.e. surface resistance set to zero). Aerodynamic conductance for momentum $g_{a,M}$ (m s$^{-1}$), following Thom (1975), for neutral conditions was derived from the regression of observed friction velocity ($u^*$, m s$^{-1}$) versus wind speed ($u$, m s$^{-1}$) measured by a sonic anemometer (Gash et al., 1999; van der Tol et al., 2003):

$$g_{a,M} = \left(\frac{u^*}{u}\right)^2 u \qquad (7)$$

It is necessary to differentiate between conductance for heat and momentum (Lankreijer et al., 1993). Following the empirical relation proposed by Garratt and Francey (1978), we used $\ln(z_{0M}/z_{0H}) = 2$ for $g_{a,H}$ (cf. Gash et al., 1999; Moors, 2012; Lankreijer et al., 1993). In addition stability corrections for non-neutral hours were implemented according to Paulson (1970). The evaporation estimated with the P-M equation is hereinafter referred to as $E_{PM-EC}$.

**3 Results**

**3.1 Rainfall**

Total rainfall ($P_G$) measured during 11 months between 19 June 2015 and 31 October 2016 (excluding the winter season from November 2015 to March 2016) was 955 mm. Mean monthly precipitation was 82 mm ($\pm$ 41 SD), with a minimum of 43 mm in May 2016 and a maximum of 156 mm in August 2015 (Fig. 2).

A total of 157 events (64 in 2015 *vs.* 93 in 2016) were identified from half hour rainfall time series during the study period. The frequency distributions of event size, duration and rainfall intensity showed a positively skewed distribution (Fig. 3). The mean (and median) event based amount, duration and intensity were 6.0 (3.1) mm, 6.5 (5.0) h and 1.1 (0.74) mm h$^{-1}$, respectively. The maximal event size, duration and intensity were 66.3 mm, 62 h and 8.5 mm h$^{-1}$, respectively.

**3.2 Throughfall, stemflow and derived interception loss**

Throughfall ($Tf$), measured for the same period as the rainfall, was 577 mm in total, corresponding to about 60 % of $P_G$. The overall standard error (SE) of $Tf$ was 48 mm (i.e. 5 % of $P_G$). No significant differences (t-test; α=0.05) were found between the mean cumulative $Tf$ estimated using either the manual array or the automated gutters, confirming that the automated





system was representative for the plot. The homogeneity in the plot was illustrated by a relatively low coefficient of variation (CV) of ~15 % (cf. Table A1).

Stemflow ($Sf$) was measured at different diametric classes from 27 July to 11 November 2016. The total $Sf$ at plot scale was 2.6 mm (1.1 % of gross rainfall). The contribution to the total $Sf$ from the four stem diametric classes, < 30 cm, 30-40 cm, 40-
50 cm and > 50 cm, was 0.8 %, 0.2 %, 0.1 % and < 0.1 %, respectively. The SE value of the overall $Sf$ was 1 % based on the accuracy of the stemflow collector.

The total interception loss estimated for the whole study period based on the wet canopy water balance was 372 mm (39 % $P_G$). The SE for the interception loss estimated as the quadratic mean of the SE's of $Tf$ and $Sf$ was 49 mm (i.e. 5 % of $P_G$).

### 3.3 Canopy related parameters

Using the mean method, we analysed data-set 1 (the calibration data-set) and found only minor differences in estimated parameter values with respect to the criteria used to select the events (Case A and B). For Case A, values of $p$ and $S$ were found of 0.32 (± 0.04) and 1.15 (± 0.25) mm, respectively. While for Case B, values of $p$ and $S$ were found of 0.28 (± 0.05) and 1.37 (± 0.51) mm, respectively (Table 4).

When we selected events based on Case A ($P_G \geq 0.5$ mm and dryness separation time of 3 h), then the linear regression
expression that relates $I$ to $P_G$ was: $I = 0.65P_G$ ($R^2 = 0.68$; $n = 24$) for non-saturated conditions and $I = 0.25P_G + 1.15$ ($R^2 = 0.67$; $n = 40$) for saturated conditions in data-set 1 (Fig. 4a). Case B (i.e. excluding events that did not reach the condition of preceding dryness validated with three fully dry LWSs) resulted in linear regressions of $I = 0.69P_G$ ($R^2 = 0.68$; $n = 11$) for unsaturated conditions ($P_G < P_G'$), and of $I = 0.23P_G + 1.37$ ($R^2 = 0.75$; $n = 17$) for saturated conditions ($P_G \geq 5$ mm) ( Fig. 4b). The derived parameters $S$ and $p$ are listed in Table 4 for all methods.
The values of the stemflow related parameters ($S_t$, $p_t$) were obtained using the method of Gash and Morton (1978) ( Fig. 4c). The trunk storage capacity ($S_t$) was estimated at 0.14 mm (±0.05) and the proportion of rain diverted to stemflow $p_t$ at 0.029 (±0.005) ($R^2 = 0.77$; $n = 12$).

Using the IEA method, average values of $p$ and $S$ of 0.22 (± 0.06) and 1.90 mm (± 0.5), respectively, were found for data-set 1 (Table 4). This result was obtained using the stricter event selection (Case B) to warrant canopy pre-dryness. Seven out of
the 17 events in data-set 1 had sufficient data points in the wetting phase to perform the respective regression analysis. One example is shown in Figure 5.

### 3.4 Energy balance closure and performance of the sonic anemometer

The energy balance closure by means of regression of $H + \lambda E$ (from eddy-covariance) versus $R_n - G_0 - Q$ for wet and dry half-hour periods was 96 %, while the energy balance ratio (EBR) defined as the sum of $H + \lambda E$ divided by $R_n - G_0 - Q$ was
0.98. The RMSE of the regression was 66.3 W m⁻² for the thirty minute interval values of $R_n$, $H$, $\lambda E$, $G$ and $Q$ (Fig. 6a).





We studied 124 half-hour periods for canopy wet conditions $P_G > 0.5$ mm for the full study period from 19 June 2015 to 31 October 2016. In addition to the eddy covariance data quality flag filtering (see Sect. 2.2.5), we tested the performance of the sonic anemometer by plotting the standard deviation of the vertical wind speed against the friction velocity. According to the Monin-Obukhov similarity theory, the ratio $\sigma w / u^*$ should be constant in neutral conditions. We found that the plot presents

a strong linear relation (Fig. 6b). The slope was consistent with previous estimations (van der Tol et al., 2003), and the offset was very close to zero.

### 3.5 Wet-canopy evaporation rates

The wet-canopy evaporation rates derived from the mean method (water balance approach) were estimated from the parameter

$\bar{E}/\bar{R}$ obtained after fitting the linear regressions. $\bar{E}/\bar{R}$ values of 0.25 ($\pm$ 0.02) and 0.23 ($\pm$ 0.03) were found for Case A and Case B, respectively. Because they were similar, we decided to use Case B (stricter event selection) to derive $\bar{E}$. The parameter $\bar{E}/\bar{R}$, multiplied by the median $R$ of 0.82 mm h$^{-1}$, results in an estimated evaporation rate of 0.19 mm h$^{-1}$.

Average micrometeorological characteristics of the wet periods are shown in Table 5. It is noteworthy that both sensible heat flux ($H$) and energy storage ($Q$) were strongly negative during wet periods. This implies a strong cooling of the surface ($Q$),

accompanied by a negative (downward) sensible heat flux. This is the case for wet periods both during the day (7AM to 7PM) and the night (7PM to 7AM). The energy balance residual, which we assume is the latent heat flux (see Eq. 4), greatly exceeds the net radiation. Other observations were: a very low vapour pressure deficit at the top of the tower, and similar average wind speed throughout the day and night (Table 5).

Between 19 June 2015 and 31 October 2016 (excluding the winter season from November 2015 to March 2016), the mean wet

evaporation rate calculated from the energy balance residual, $E_{EB-EC}$, was 0.28 mm h$^{-1}$ during the day (Fig. 8a), 0.07 mm h$^{-1}$ during the night (Fig. 8b), and 0.20 mm h$^{-1}$ for day and night periods combined (Fig. 8c). The main sources of evaporation heat (in equivalent water depth unit) were: net radiation (0.07 mm h$^{-1}$), sensible heat (0.09 mm h$^{-1}$) and the release of stored energy within the canopy (0.03 mm h$^{-1}$). Evaporation rates derived from the energy balance residual (for day and night periods) presented a skewed distribution with values ranging from -0.53 to 2.59 (mm h$^{-1}$) with a median value of 0.13 (mm h$^{-1}$).

In order to estimate average wet canopy evaporation with the Penman-Monteith equation, we first estimated aerodynamic conductance to momentum $g_{a,M}$ (m s$^{-1}$). We estimated the aerodynamic conductance to momentum for the predominant SW wind direction and selected the fluxes coming from the wind direction between 180° to 360°. In this direction, effects of the tower construction on the wind were minimal. The $g_{a,M,EC}$ was estimated by means of the regression between wind speed and friction velocity as $g_{a,M,EC} = 0.0318\, u$ (Fig. 7). When we applied the stability correction for non-neutral hours (Paulson, 1970)

and used $\ln(z_{0M}/z_{0H}) = 2$ (Lankreijer et al., 1993; Moors, 2012), we obtained an aerodynamic conductance to water vapour as: $g_{a,H,EC} = 0.0303\, u$.





For the whole study period, using the estimated $g_{a,H,EC}$, and considering $Q$ and $G$ to be part of the available energy ($A = Rn + Q + G$), the mean and median wet evaporation rates estimated by the Penman-Monteith equation ($E_{PM-EC}$) were 0.13 and 0.10 mm h$^{-1}$, respectively (Figure 8 f, Table 6). The period 19 June – 31 October 2015 presented similar mean and median evaporation rates to the values estimated for the period 1 April – 31 October 2016 (Table 6).

In general, the mean $E$ estimated with the P-M equation (using $g_{a,H,EC} = 0.0303 \, u$) was 35 % lower than the mean $E$ derived from the energy balance residual. Using the water balance measurements with the mean method resulted in an estimated evaporation rate of 0.19 mm h$^{-1}$, which is similar to the mean values of $E_{EB-EC}$, although 40 % higher than the estimated values of the median $E_{EB-EC}$ used in the Gash model (Table 6).

**3.6 Modelling rainfall interception**

We used two different parametrizations of the Gash model. In the first parametrization (Run 1) the parameters $S$, $p$ and $\bar{E}/\bar{R}$ were derived by using the mean method. In the second parametrization (Run 2), the parameters $S$ and $p$ were derived from the IEA, the parameter $\bar{E}/\bar{R}$ was calculated with $\bar{E}$ as the median $E$ derived from the energy balance residual and $\bar{R}$ as the median rainfall intensity derived from the tipping bucket measurements (Table 7).

Run 1 underestimated the predicted interception loss by 3 % with a RMSE of 0.93 mm for the calibration data set (Table 7 and

Fig. 9). The model performance based on the Nash-Sutcliffe model efficiency (NSE) was very good (0.90) (Table 7). Run 2, which used the median value of $E_{EB-EC}$ (0.12 mm h$^{-1}$) and the median $R$ (0.82 mm h$^{-1}$) to obtain parameter $\bar{E}/\bar{R}$ (0.15), underestimated $I$ by about 8 % (166.6 mm modelled, versus 180.4 measured $I$). The RMSE was 1.36 mm and the performance was lower than that of Run 1 (NSE = 0.79). The predicted total interception loss for the validation data-set using both Run 1V (V for validation data-set) and Run 2V configurations were in good agreement with $I$ derived from the $Tf$ and $Sf$

measurements, with relative errors of about 1 %. In both cases the RMSE was about 0.79 mm and the model performance was good with a NSE of 0.79.

The interception components that contributed most to the overall evaporation interception loss differ between the two parametrizations of model validation, Run 1V and Run 2V. In the first case, the two largest contributors were evaporation from the saturated canopy during rainfall (37 %) and evaporation loss during the drying phase (34 %). In Run 2, the same two

components were the greatest contributors, but in opposite order: evaporation loss during the drying phase was the main contributor (44%), and evaporation from the saturated canopy during rainfall was the second contributor (24 %). The third largest contribution, in both cases (Run 1 and Run 2), was evaporation from small rainfall events (18 % and 24 %, respectively), followed by the contribution from the wetting phase (7 % and 5 %, respectively). Evaporation from trunks (during saturated and unsaturated conditions) contributed less than 5 % for both data-sets (Table 8).

The sensitivity of the Gash model results to changes in $S$ and $\bar{E}/\bar{R}$ are shown in Fig. 10. The contour lines represent the domain of RMSE for different combinations of parameters. Figure 10 shows that parameter equifinality (Beven, 1993) occurs between $S$ and $\bar{E}/\bar{R}$ (van Dijk et al., 2015), which implies in this case that an underestimation of $S$ is likely to be compensated by





overestimation of $\bar{E}/\bar{R}$. This effect can be seen when modelling the validation data-set (Fig. 10b): for Run 1 a low value of $S$ (1.37 mm) may be compensated by a high value of $\bar{E}/\bar{R}$ (0.23), leading to a similar RMSE as for Run 2, which used a higher value of $S$ (1.90 mm) and a lower value of $\bar{E}/\bar{R}$ (0.15). A similar effect is detected when modelling data-set 1 (Fig. 10a): both parametrizations (Run1 and Run 2) produce a relative error lower than 10 % (Table 7).

**4 Discussion**

**4.1 Canopy storage capacity**

In the same study area, Klaassen et al. (1998) evaluated the most common indirect methods to derive $S$ from multi-event throughfall measurements. They found that the mean method tended to underestimate $S$ compared to direct microwave transmission measuring. However, indirect methods are still largely used due to their low cost and simplicity. As an alternative

to the multi-event methods, Link et al. (2004) proposed analysis of individual events. They found that the assumption of a constant $\bar{E}/\bar{R}$ during multiple events was unsustainable, especially during the wetting phase, and might contribute to the underestimation of $S$ in the mean method.

Our findings confirm that $S$ estimated with the mean method is lower (-30 % lower) than $S$ estimated with the IEA. To avoid underestimation caused by incorporating events not preceded by canopy dryness in the regression analysis, we made use of

wetness sensors. However, our findings indicate that, despite using the wetness sensors to eliminate certain events, the results remained similar (Table 4).

The storage capacity at the study site had been reduced compared to in earlier studies, very likely due to the decrease in tree density and LAI over the years. When the stand was 29 years old, the tree density was 992 trees ha$^{-1}$ and LAI was 8 ( Table 2), while Klaassen et al. (1998) used the principle of microwave attenuation to determine an $S$ of 2.4 mm. At the time of the

present study, the tree density as well as the LAI had decreased by about 40 % (Table 2). However, the average total storage capacity ($S + S_t$) for the study period was reduced much less, only by about 20 %, if we considered $S$ (2.0 mm) derived with the IEA method.

Our estimation of $S$ is comparable with that of other old Douglas-fir stands (Table 9) and supports the hypothesis that LAI might not be the main predictor of $S$ for Douglas-fir forests under temperate climatic conditions. Rutter et al. (1975) reported

a total storage capacity of 2.1 ($S$ equal to 1.2 mm and $St$ of 0.9 mm) for Bramshill Forest (UK), a 42-years-old Douglas-fir stand with similar density but larger LAI (660 trees ha$^{-1}$, LAI = 12). In contrast, Link et al. (2004) applied IEA in a 500-year-old mixed Douglas-fir and Western hemlock forest (560 trees ha$^{-1}$, LAI = 8.6) located in Washington (USA), and found larger values of $S$ ranging from 2.71 mm to 4.17 mm. Pypker et al. (2005) in South Central Washington found significant differences in $S$ for a young (25-year-old) Douglas-fir forest and an old-growth (>450-year-old ) mixed Douglas-fir –and Western hemlock

forest, with $S$-young being 1.4 mm; and $S$-old-growth being 3.3 mm. Both forests had a similar LAI (LAI-young = 10.2; LAI-old-growth = 9.6), however, despite a notable difference in stem the density (young: 2200 tree ha$^{-1}$, old-growth: 441 tree ha$^{-1}$),





the larger $S$ was found in the old-growth forest. The high $S$ was presumably caused by the changes in species composition (i.e. presence of understory) and the presence of epiphytes, conditions that were not observed at our study site.

Some authors have found that $S$ can be linearly related to the fraction of vegetation cover, which implies an exponential relation between $S$ and LAI (Moors, 2012). However, for closed canopies (LAI > 5) in temperate climate utilizing the fraction of vegetation cover might not be an option (Moors, 2012). The relation between $S$ and the number of trees and their basal area has also been noted to be valid for several sites (Turner and Lambert, 1987; Teklehaimanot et al., 1991). We speculate that for pine species the tree density and basal area imply a direct relation with the amount of bark present in the forest. This amount will increase as the canopy gets older and taller. Recent research in cedar trees in Japan has found high values of $S$, and has shown the bark on the stems providing a major contribution (Iida et al., 2017). We only found a value of $S_t$ of 0.14, but part of the trunks' storage capacity may be hidden in the estimated $S$.

Our results suggest that a long-term decrease in $S$ in a Douglas-fir stand not necessarily implies a decrease in $I$, because the natural process of tree growth as well as forest management practices, such as thinning, all affect other variables that influence the rainfall interception process. Direct extrapolation of $I$ by means of LAI without considering other driving forces (i.e. aerodynamic conductance or change in energy storage) can lead to erroneous approximations of interception loss.

## 4.2 Wet canopy evaporation rate

Previous investigations have shown that it is possible to estimate sensible heat flux from the sonic anemometer during rainy conditions. Latent heat fluxes derived from the energy balance residual have been shown to be a good approach to derive evaporation rates during rainfall. However, discrepancies with the water budget approach and the Penman-Monteith equation have been reported in several studies (Ringgaard et al., 2014; Schellekens et al., 1999; Holwerda et al., 2012).

The value of $\bar{E} = 0.20$ mm h$^{-1}$ from the present study was derived from the energy balance residual and is in agreement with the canopy structural changes that occurred in Speulderbos due to natural growth and to management practices causing reduced density. At the Speulderbos study site, when the stand was 29 years old (when canopy height was 18 m and LAI was 8 m$^2$ m$^{-2}$) Klaassen et al. (1998) used a combination of eddy correlation and psychrometer profile measurements, and reported a wet canopy evaporation rate of 0.077 mm h$^{-1}$ (55 W m$^{-2}$). This value was lower than evaporation rates derived at other coniferous forests of the same height and stand configuration and in similar climatic conditions. For example, in the Hafren forest (United Kingdom) Stewart (1977) used the Bowen ratio method and found a value of 0.19 mm h$^{-1}$ from day-time measurements, while Gash et al. (1980) used the Penman-Monteith equation and found a similar evaporation rate of 0.13 mm h$^{-1}$. Link et al. (2004) reported an average evaporation rate of 0.14 mm h$^{-1}$ using the P-M equation in a 500-year-old mixed Douglas-fir forest (60 m height) (Table 9). In a mixed plantation in west Denmark, Ringgaard et al. (2014) found a wet canopy evaporation rate of 0.21 mm h$^{-1}$ during the summer season.

In a long-term comparison done by Pypker et al. (2005), similar values of wet evaporation rates were found (young forest: 0.25 mm h$^{-1}$, old-growth forest: 0.21 mm h$^{-1}$). However, the studied old-growth forest was a mixture of Douglas-fir and Western hemlock with understory present, and the young stand had a very high tree density (Table 9). Our study site offered the



advantage of an unchanged species composition, which allowed us to focus on the long-term effects of the changes in tree density and LAI. Over the past 25 years, tree density and LAI at our study site mainly declined as the result of thinning practices, and to a lesser degree due to natural tree mortality. Moreover, forest height increased by about 16 m. The combination of these changes resulted in an increase in aerodynamic conductance from 0.065 m s$^{-1}$ to 0.105 m s$^{-1}$. This change has a direct

impact on the exchange of fluxes between canopy and the atmosphere (Holwerda et al., 2012; Schellekens et al., 1999; Moors, 2012).

$\bar{E}$ (mean) estimated by means of the Penman-Monteith equation was about 35 % lower than with the energy balance residual approach. Several explanations have been proposed in the literature to explain such discrepancies in similar studies. In a comprehensive analysis, van Dijk et al. (2015) pointed out the following possible reasons: i) underestimation of biomass and

heat ground release; ii) underestimation of aerodynamic conductance; iii) unaccounted energy advection; iv) errors in air humidity measurements; v) mechanical water transport. In our analysis, we have incorporated estimations of $Q$ and $G$ in our estimate of the available energy $A$ in the P-M equation, and therefore disregard the underestimation of biomass and heat ground release as main cause of the underestimation of $E$. In the estimation of the aerodynamic conductance, we used $g_{a,\mathrm{H}}$, estimated using friction velocity derived from the eddy covariance system, corrected for stability (Paulson, 1970). In 2015, van Dijk et

al. pointed out that $g_{a,\mathrm{H}}$ and $g_{a,M}$ require the validity of the Monin–Obukhov similarity theory (MOST). MOST assumes that the turbulent flow ($u^*$) is the only important velocity scale, and that the height above zero-plane of displacement ($z - d$) and Obukhov length ($L$) are the only important length scales (van Dijk et al., 2015). In the case of tall canopies, as at the Speulderbos site, the observations are taken in the roughness sub-layer. In this layer, the turbulence is also influenced by length scales related to the surface (i.e. mixing layer instability at the top of the canopy). Corrections for this effect in MOST during

rainfall have not been developed and could lead to an underestimation of aerodynamic conductance. Energy advection has also been hypothesized to be a source of unaccounted additional energy (Shuttleworth and Calder, 1979). Although it appeared to occur mainly at maritime sites (Schellekens et al., 1999), Stewart (1977) advocated that the energy does not necessarily need to come from the ocean, and could be supplied by drier and warmer nearby areas. Although vertical energy advection would not invalidate the P-M equation, horizontally advected energy occurring below the level of energy balance measurements

would not be accounted for, and could influence the underestimation of $E$. Our results suggest that vertical sensible heat flux measured as negative $H$ is the main source of energy that sustains $E$ during rainfall (Table 5), however this situation was not predicted by the P-M equation. This could be attributed to errors in air humidity measurements. Wallace and McJannet (2008) estimated that 2 % overestimation in RH already leads to an underestimation of $E_{PM}$ of 36 %. This situation may occur in our case as the accuracy of our RH sensor (CS215, Campbell Sci. Inc.) during wet conditions (> 90 % RH) is low (± 4 %). Given

the low rainfall intensities prevailing in the study area, the effect of splash droplet evaporation (Murakami, 2006) was not included in the present study.





### 4.3 Rainfall interception

The interception loss derived for the two consecutive growing seasons of 2015 and 2016 was 37 % and 39 % of gross rainfall ($P_G$), respectively. These values are in agreement with other similar studies (Table 9). For similar climatic conditions, Rutter et al. (1971) investigated a Douglas-fir stand of similar age and stem density in Bramshill Forest (UK) and found an $I$ of 39 %

of $P_G$. Soubie et al. (2016) found a value for $I$ of 30 % in a Douglas-fir stand in Belgium, which is lower than the value found in the current study and may be attributed to the difference in stem density (145 tree ha$^{-1}$) since LAI was similar (LAI = 4.2). In a comparison between mixed young and old Douglas-fir stands in the North West Pacific (Washington, US), Pypker et al. (2005) reported similar values of $I$, namely 21 % and 24 %, respectively. They attributed the slightly larger $I$ in the old stand to the higher $S$, which was also linked to the change in species composition and to the presence of epiphytes.

The other interesting finding in the study by Pypker et al. (2015) is that $\bar{E}/\bar{R}$ was similar for both stands (i.e. young and old). Because the older stand was taller, the aerodynamic conductance may have been larger, with larger expected evaporative rates as a consequence (Teklehaimanot et al., 1991). In contrary, and considering that $\bar{R}$ was not variable (the two stands were close to each other), the evaporative fluxes from wet canopy were similar. Pypker et al. (2005) observed that the variable species composition at their study site (i.e. Douglas-fir mixed with Western hemlocks) increased the gap size and influence on $g_a$.

They explained that in this particular situation the use of the average Douglas-fir height was likely inappropriate for calculating $d$ and $z_0$ and hence $\bar{E}/\bar{R}$.

In the case of Speulderbos at a younger stage, the larger stem density and higher LAI meant a larger $S$ (Klaassen et al., 1998), while, at an older stage (2015-2016), the $\bar{E}/\bar{R}$ was larger mainly as effect of a larger $g_a$, due to the larger canopy height in combination with lower tree density (Teklehaimanot et al., 1991).

The original version of the Gash (1979) analytical model successfully predicted $I$ for the calibration and validation data-sets (Table 7). Several studies have demonstrated the validity of the Gash model in temperate coniferous forests (Muzylo et al., 2009). However, one of the pitfalls of the Gash model is the parameter equifinality between $\bar{E}$ and $S$ (van Dijk et al., 2015; Klaassen et al., 1998) which means that an error in the estimation of $S$ (i.e. underestimation) can compensate an error in $\bar{E}$ (i.e. overestimation).

The performance of the Gash model was as good as that of some more sophisticated models applied in earlier studies at the same site (i.e. Bouten et al., 1996). The difference between observed and predicted values of $I$ was lower than in previous applications of the Gash model in similar climatic conditions (Lankreijer et al., 1999). The model overestimated the total $I$ for all parametrizations (Table 7). During the calibration the relative error of total $I$ was lower than 8 %, and for the validation phase it was lower than 2 %.

We used the RMSE as a criterion to evaluate the sensitivity of the model to the two main parameters in the Gash model, $S$ and $\bar{E}/\bar{R}$, considering that the other independently derived parameters have less influence on the model and can be kept fixed. We observed that for calibration (Fig. 10a) and validation (Fig. 10b) certain combinations of $S$ and $\bar{E}/\bar{R}$ yield an almost equally low RMSE. Such combinations along the major axis of the inner elipse that embraces the optimal solution represent the above



mentioned functional equivalence between $S$ and $\bar{E}/\bar{R}$. For instance, in Fig. 10b, parametrization for Run 1 presents a higher $\bar{E}/\bar{R}$ (and lower $S$) that the one set for Run 2, with a similar RMSE of below 0.8 mm. This confirms that it is necessary to reduce the uncertainty in one of the parameters ($S$ or $\bar{E}$ ) by independent measurements, before optimizing the second one (i.e. Gash et al., 1980; Ringgaard et al., 2014). The independent estimations of $S$ and $\bar{E}$ by means of the IEA and the energy balance

residual give us confidence to select the parametrization of Run 2 as the most realistic in the case of Speulderbos.

## 5 Conclusion

Rainfall interception loss ($I$) was quantified and modelled for a mature Douglas-fir stand (ca. 55 years old) in the centre of the Netherlands for two growing seasons in 2015 and 2016. Over the study period the interception loss was 38 % of gross precipitation. This value was similar to the value reported for the same stand when the forest was 29 years old, indicating the

changes in forest structure may not always result in changes in interception loss. Tree density as well as LAI were reduced by about 40 % in comparison with the former study by Klaassen et al. (1998). However, the change in canopy storage capacity ($S$) was much less (a reduction of about 20 %). Canopy storage capacity remained relatively stable largely due to the increase of the total biomass, and more specifically of stem bark surface. The reduction in stem density and the growth of canopy height resulted in a larger surface roughness and in consequence enhanced the evaporation rate during rainfall.

The main sources of energy supply that sustain evaporation of intercepted rainfall were net radiation (35%), sensible heat flux (45%) and change in energy stored in air and canopy biomass (15%). Downward sensible heat fluxes estimated by means of the eddy covariance technique were larger than those predicted by the P-M equation, possibly due to inaccuracies in the measurement of the relative humidity in the air.

The Gash model was able to simulate $I$ reasonably well, with relative errors of less than 10 %. A sensitivity analysis of

interception losses simulated with the Gash model shows that the presently higher $\bar{E}/\bar{R}$ can indeed compensate for the lower $S$, confirming the parameter equifinality effect.

Our results confirm that even after a reduction in tree densities old growth stands can maintain similarly high rates of interception. This finding will be useful to improve long-term model predictions that involve structural changes or planned management practices in forested ecosystems.

## Acknowledgements

The authors thank Myriam Miranda, and Murat Ucer for their help with the field measurements. The authors also thank Peiqi Yang for his support in writing the Matlab code. Cesar Cisneros Vaca has received funding from the Secretariat for Science and Technology of Ecuador (SENESCYT), Contract No. 2013-AR2Q2741.





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





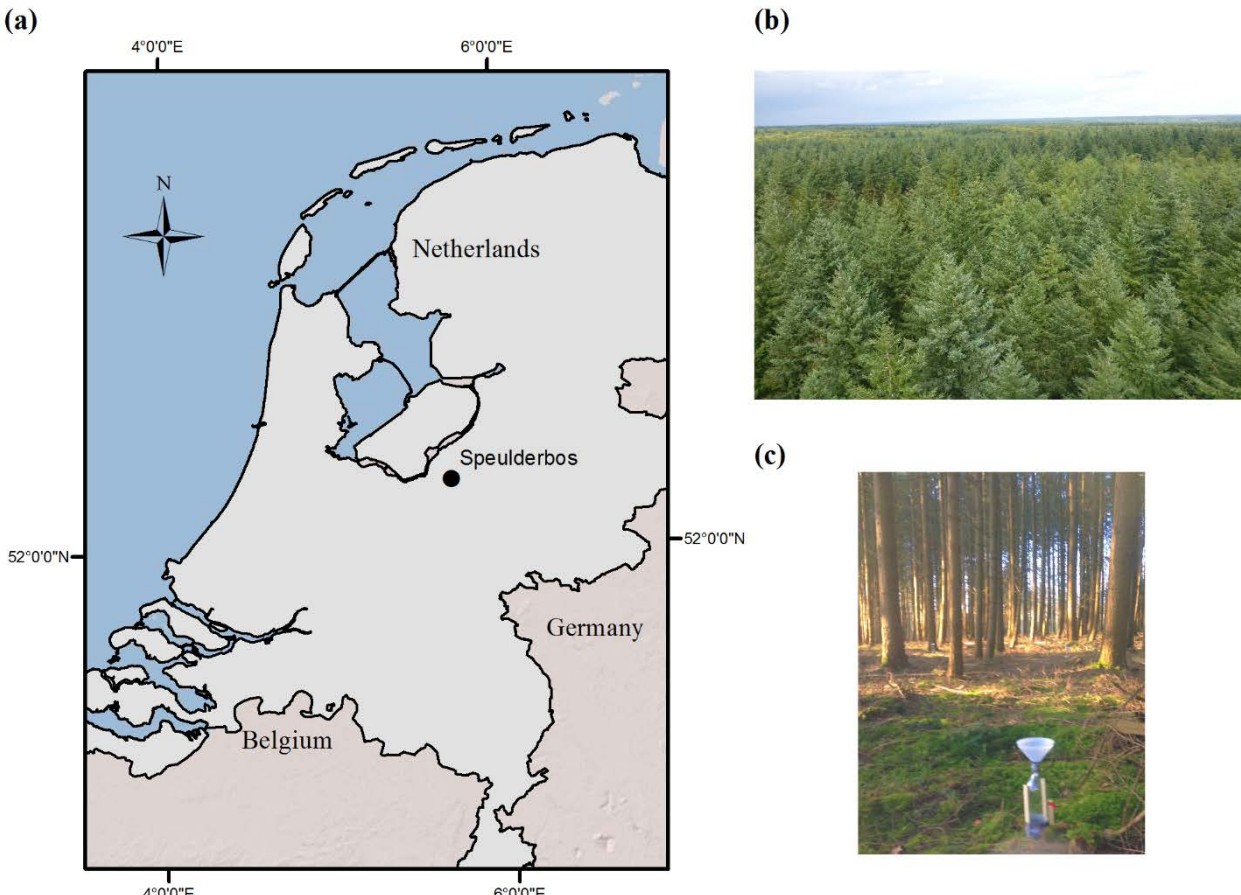

**Figure 1. Study area in Speulderbos: (a) Location map in the Netherlands; (b) top view of the canopy; (c) funnel-type collector used to quantify throughfall in the study site.**



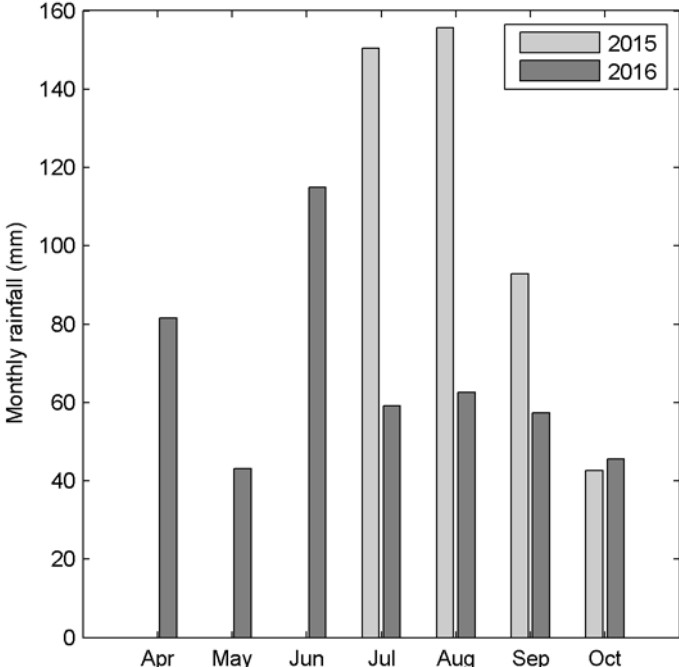

**Figure 2. Monthly rainfall at the study area (Speulderbos) for the study period from 19 June 2015 to 31 October 2015 and from 1 April 2016 to 31 October 2016.**





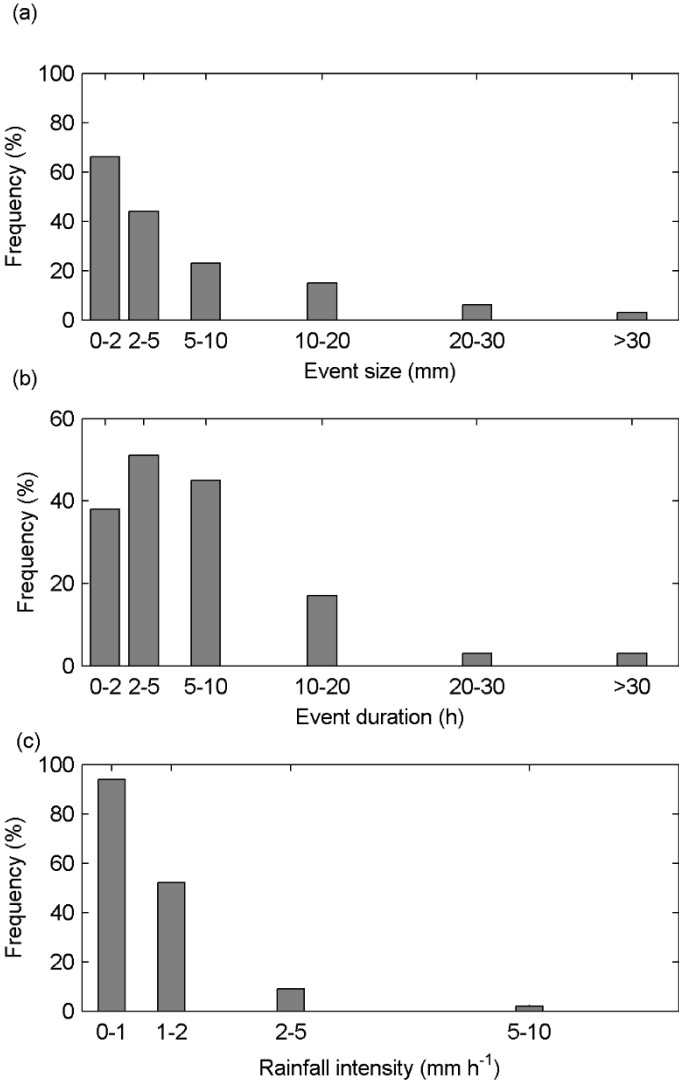

**Figure 3. Frequency distributions of: (a) amount, (b) duration and (c) intensity of rainfall events during the study period.**



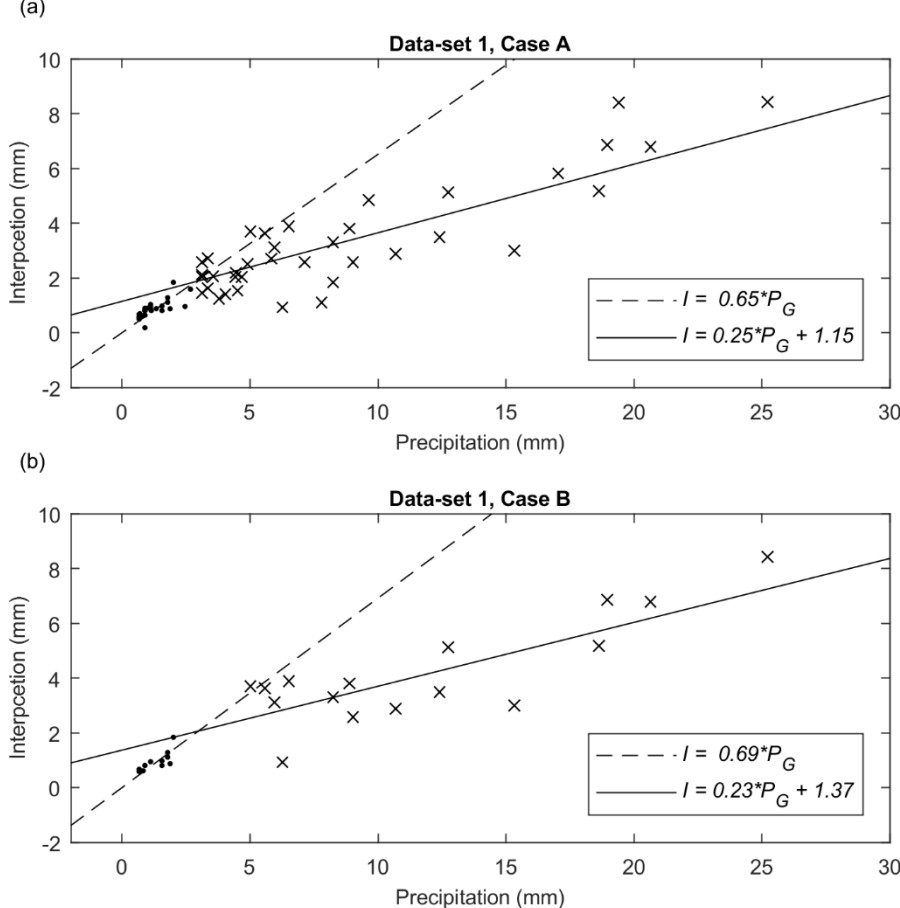

**Figure 4. Determination of canopy related parameters using the mean method. Dots represent rainfall events with total rainfall less than the necessary for saturation, crosses represent data with enough rainfall to saturate the canopy. (a) Linear regression using data-set 1, events selected in Case A. (b) Linear regression using data-set 1, events selected in Case B.**





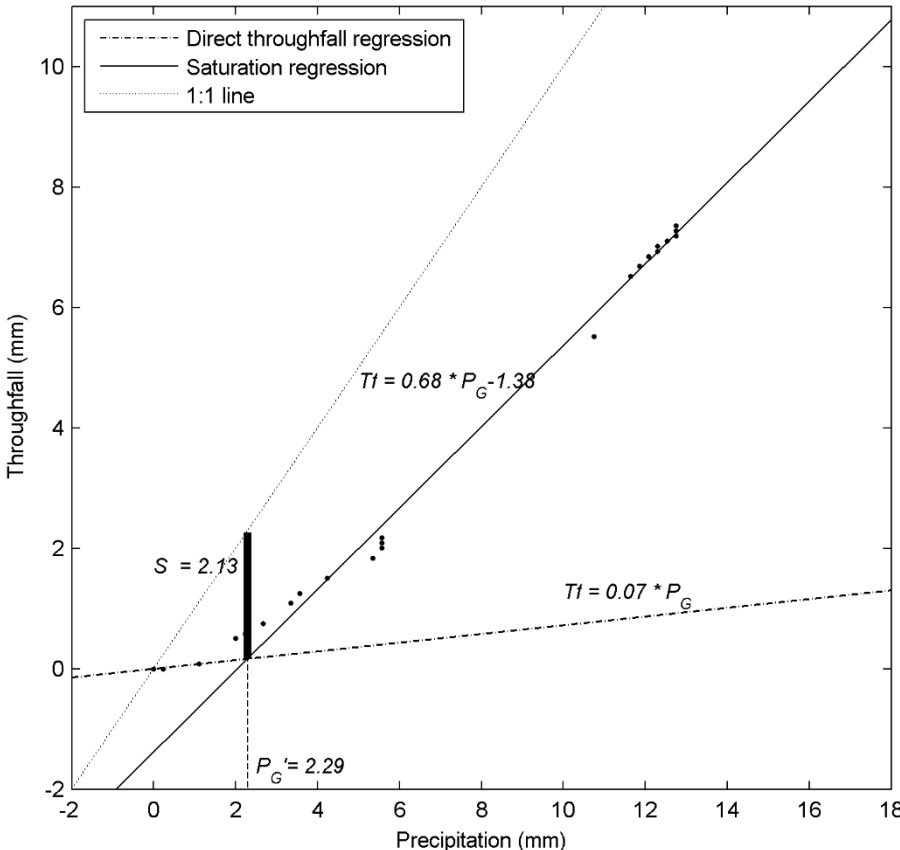

**Figure 5. Example of individual event analysis (IEA) on 17 September 2015, plot of data used to estimate canopy direct throughfall and saturation storage capacity, assuming no evaporation during the wetting phase. Dots represent values of cumulative rainfall versus cumulative $Tf$. The direct throughfall regression equation for this particular event was $Tf = 0.07\, P_G$. The saturation regression equation was $Tf = 0.68\, P_G - 1.38$. Canopy saturation point ($P_G'$) was calculated as the intersection of the two linear regressions, $P_G' = 2.19$ mm.**





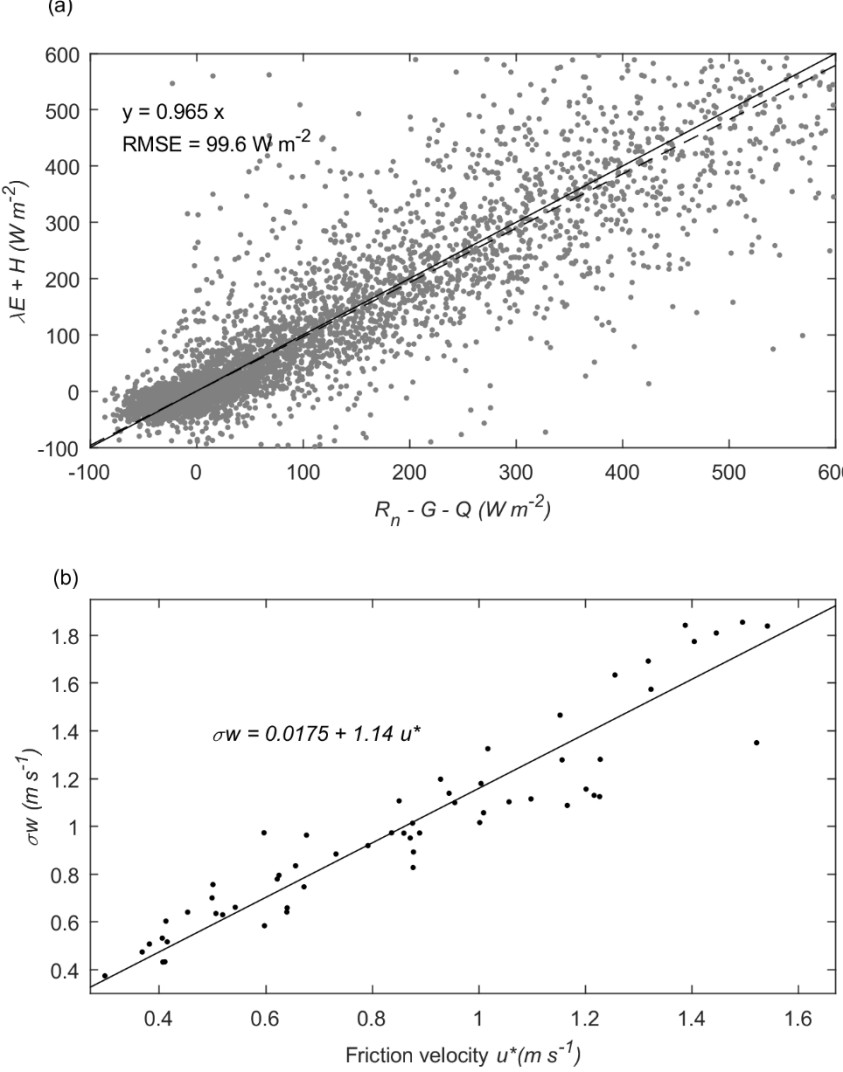

**Figure 6. (a) Half-hour interval turbulent heat fluxes ($H$ and $\lambda E$) versus available energy ($R_n - G - Q$) for the study site. Solid line represents 1:1 line and dashed line represents linear regression forced through the origin. (b) Half-hour averages of standard deviation of the vertical wind speed $\sigma w$ (m s$^{-1}$) versus friction velocity $u^*$ (m s$^{-1}$), wet canopy conditions $P_G > 0.5$ mm (30 min)$^{-1}$, and near neutral stability (-0.02 < $(z - d)/L$ < 0.02).**





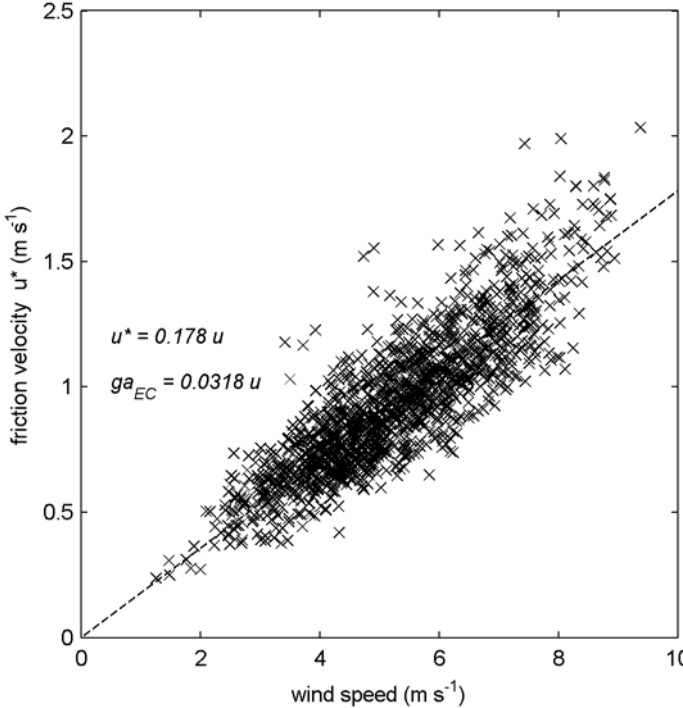

**Figure 7. Linear regression of friction velocity *u\** against wind speed *u* for near neutral hours (-0.02 < (*z − d*)/*L* <0.02) and from wind SW direction.**




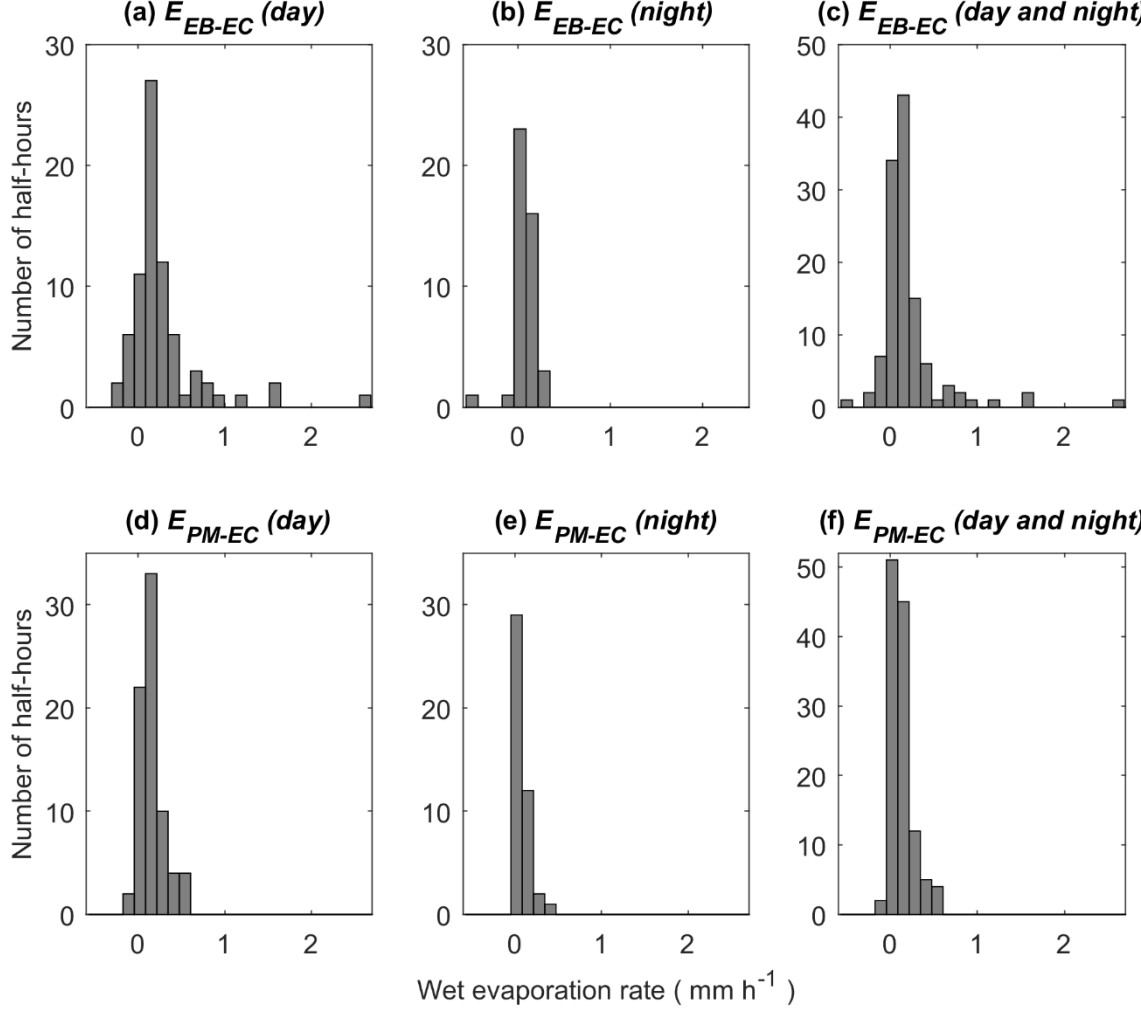

**Figure 8. Distributions of wet canopy evaporation rates during day time (7:00-19:00 LT), night time (19:00-7:00 LT) and combined day and night. Two different methods applied: (a-c) energy balance residual ($E_{EB-EC}$) and (d-f) Penman-Monteith ($E_{PM-EC}$).**





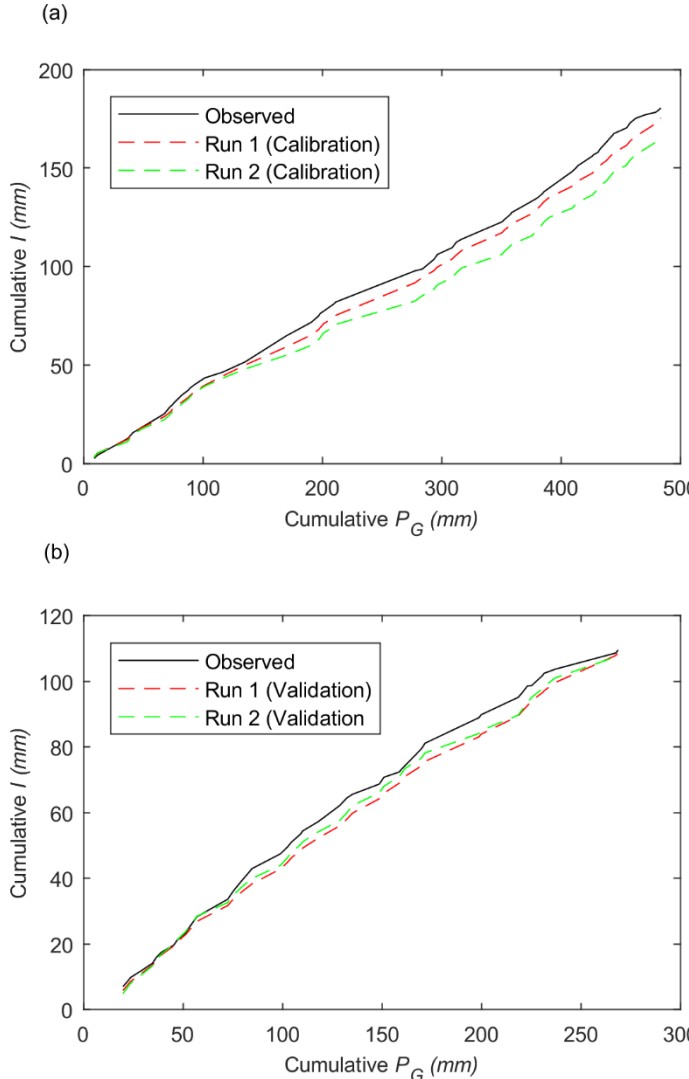

**Figure 9. Cumulative measured and modelled interception loss for (a) two parametrizations of the Gash model, Run 1 and Run 2, using the calibration data-set 1, (b) Validation of two parametrizations of the Gash model using the validation data-set 2.**





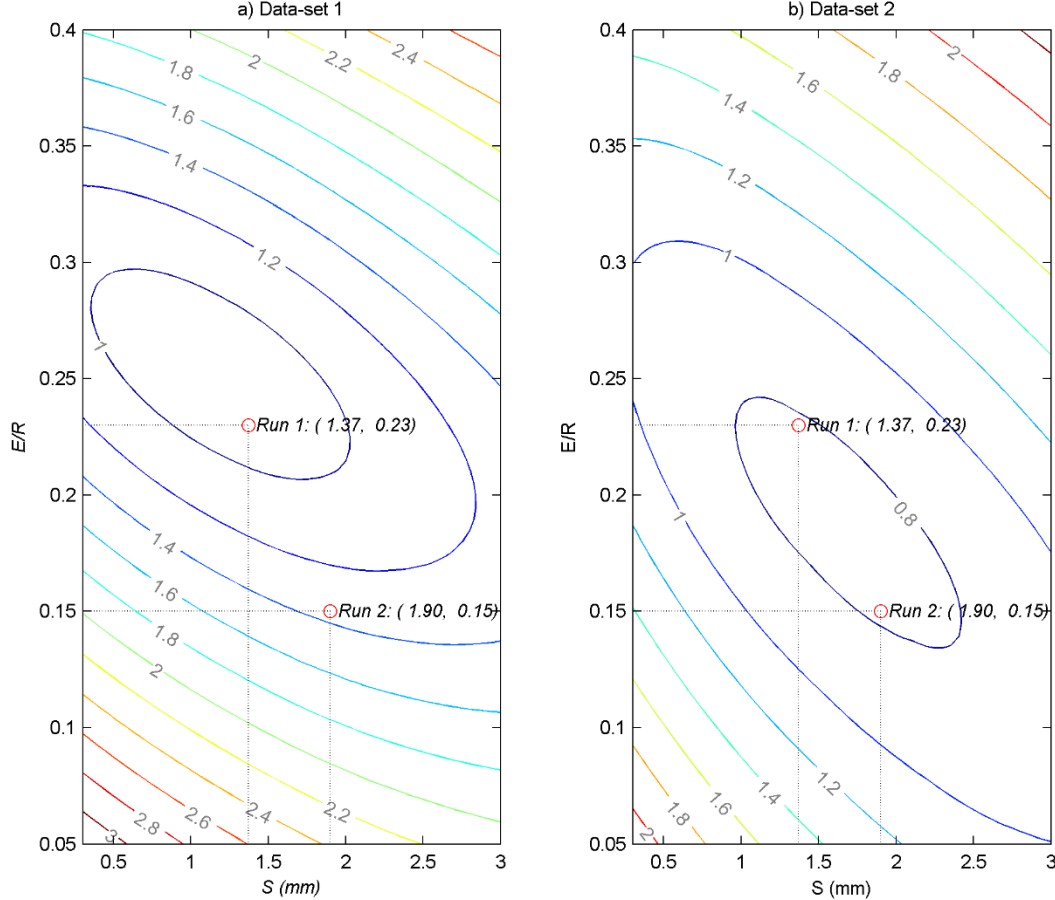

**Figure 10. Sensitivity analysis of the parametrized original Gash model. Contour lines representing the RMSE for different combinations of the parameters' canopy storage capacity (S) and the ratio $\bar{E}/\bar{R}$. (a) Sensitivity analysis using calibration data–set 1 (19 June 2015 to 31 Oct 2015. (b) Sensitivity analysis using validation data-set 2 (19 June 2016 to 31 Oct 2016). The red circles represent the corresponding parameters used in the model Run 1 and Run 2.**





**Table 1. Main micro-meteorological instruments installed on the Speulderbos flux tower.**

| Data-logger | Instruments | Parameters | Height (m) |
|---|---|---|---|
| CR5000 | Sonic anemometer CSAT3 (Campbell Sci. Inc.) | Wind speed 3D components ($u$, $v$, $w$), Sonic temperature | 47 |
| | LI7500 gas analyser (Li-Cor Biosciences) | Water vapour and $CO_2$ concentrations | 47 |
| CR23X_1 | Net radiometer CNR1 (Kipp and Zonen) | Four-components on net radiation and Temperature | 35 |
| | 3 Leaf wetness sensor Model 237 (Campbell Sci. Inc.) | Leaf wetness sensor, wetness status | 26, 24, 20 |
| CR1000 | Temperature and humidity sensor CS215 (Campbell Sci. Inc.) | Air temperature, relative humidity | 46 |
| | Temperature and humidity sensor CS215 (Campbell Sci. Inc.) | Air temperature, relative humidity | 38 |
| | Temperature and humidity sensor HC2-S3C03 (Rotronic) | Air temperature, relative humidity | 32 |
| | Temperature and humidity sensor HC2-S3C03 (Rotronic) | Air temperature, relative humidity | 24 |
| | Temperature and humidity sensor HC2-S3C03 (Rotronic) | Air temperature, relative humidity | 16 |
| | Temperature and humidity sensor HC2-S3C03 (Rotronic) | Air temperature, relative humidity | 4 |
| CR23X_2 | Barometer (Campbell Sci. Inc.) | Air pressure | 1 |
| | Two soil heat flux plates HFP01 (Hukseflux) | Soil heat flux | -0.08 |





**Table 2. Comparison of stand parameters and biomass dry weight (DW) for Douglas-fir stand in Speulderbos. Above ground biomass determined by means of stem survey and allometric relationships from Bartelink (1996).**

| Data-logger | 2015 | 1988[a] |
|---|---|---|
| Tree density (number ha$^{-1}$) | 571 | 992 |
| Mean DBH (cm) | 34.8 | 20.7 |
| LAI[b] (m$^2$ m$^{-2}$) | 4.5 | 8[c] |
| Stem wood DW (kg m$^{-2}$) | 29.9 | 14.6 |
| Branches DW (kg m$^{-2}$) | 1.9 | 0.9 |
| Needles DW (kg m-2) | 1.2 | 0.8 |
| Total biomass DW (kg m$^{-2}$) | 33.18 | 16.3 |

[a] Biomass dry weight values were estimated using tree density and DBH from Tiktak and Bouten (1994)

[b] LAI measured by using Licor-2000 instrument.

[c] Previous studies in Speulderbos reports a LAI of 11 that value was estimated by the destructive method (cf. Heij and Schneider, 1991). For comparative reasons we use the reported value using the Li-Cor photometer.

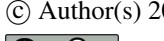



**Table 3. Main equations of the analytical Gash (1979) interception model.**

| Component of the model | Formulation |
|---|---|
| For $m$ storms with $P_G$ insufficient to saturate the canopy ($P_G < P_G'$) | $(1-p-p_t)\sum_{j=1}^{m} P_{G,j}$ |
| Wetting up the canopy with n storms large enough to saturate the canopy ($P_G \geq P_G'$) | $n(1-p-p_t)P_G'-nS$ |
| Evaporation from the saturated canopy during rainfall | $\overline{E}/\overline{R}\sum_{j=1}^{n}(P_{G,j}-P_G')$ |
| Evaporation after rainfall event | $nS$ |
| Evaporation from trunks for $q$ storms large enough to saturate trunk storage ($P_G \geq S_t/p_t$) | $qS_t$ |
| Evaporation from trunks for small storms unable to saturate the trunk storage ($P_G < S_t/p_t$) | $p_t \sum_{j=1}^{m+n-q} P_{G,j}$ |





**Table 4. Forest structure parameters: canopy storage capacity ($S$), free throughfall coefficient ($p$), ratio average evaporation over average rainfall intensity ($\overline{E}/\overline{R}$), derived by using two different methods: mean method, and individual event analysis (IEA).**

| Method | $S$ (mm) | | $p$ | | $\overline{E}/\overline{R}$ (±SE) | |
|---|---|---|---|---|---|---|
| | Case A[a] | Case B[b] | Case A | Case B | Case A | Case B |
| Mean method | 1.15(±0.25) | 1.37 (±0.51) | 0.32 (±0.04) | 0.28 (±0.05) | 0.25 (±0.02) | 0.23 (±0.03) |
| IEA | - | 1.90 (± 0.5) | - | 0.17 (±0.06) | - | - |

[a] Case A: Event selection based on amount of $P_G$ and using a pre-dry period of 3 h.

[b] Case B: Events with pre-dry period where two leaf wetness sensors (LWS) indicating fully dry canopy. Saturated condition $P_G > 5$ mm.





**Table 5. Average micro-meteorological characteristics for half-hour periods with more than 0.25 mm (30 min)$^{-1}$ of $P_G$ for day (7:00—19:00 LT) and night conditions (19:00-7:00 LT)**.

| Parameter | Day ($n$ = 75) (±SD) | Night ($n$ = 44) (±SD) | Day and Night ($n$ = 119) (±SD) |
|---|---|---|---|
| Net Radiation (W m$^{-2}$) | 78 (±95) | -2 (±12) | 48 (±85) |
| Sensible heat flux (W m$^{-2}$) | -94 (±253) | -25 (±77) | -66 (±204) |
| Total energy storage rate (W m$^{-2}$) | -24 (±56) | -15 (±32) | -20 (±48) |
| Soil heat flux (W m$^{-2}$) | 0.5 (±3.2) | -2 (±3.6) | -0.4 (±3.6) |
| Air temperature (°C) | 12.7 (±4) | 11.8 (±5) | 12 (±5) |
| Vapour pressure deficit (hPa) | 0.7 (±0.8) | 0.6 (±1.4) | 0.7 (±0.8) |
| Wind speed (m s$^{-1}$) | 3.7 (±1.7) | 3.2 (±1.0) | 3.5 (±1.4) |



**Table 6. Summary statistics for the wet evaporation rates estimated for the study period by different methods: energy balance($\overline{E}_{EB-EC}$), and Penman-Monteith equation ($\overline{E}_{PM-EC}$).**

| Method | Energy balance residual | | | Penman-Monteith | | |
|---|---|---|---|---|---|---|
| | $\overline{E}_{EB-EC}$ (mm h$^{-1}$) | | | $\overline{E}_{PM-EC}$ (mm h$^{-1}$) | | |
| | Jun-Oct 2015 ($n = 68$) | Apr-Oct 2016 ($n = 51$) | All[a] ($n = 119$) | Jun-Oct 2015 ($n = 68$) | Apr-Oct 2016 ($n = 51$) | All ($n = 119$) |
| Mean | 0.23, | 0.16 | 0.20 | 0.12 | 0.13 | 0.13 |
| Median | 0.12, | 0.12 | 0.12 | 0.10 | 0.10 | 0.10 |
| Range | [-0.53, 2.59] | [-0.26, 0.75] | [-0.53, 2.59] | [-0.01, 0.50] | [-0.06, 0.57] | [-0.06, 0.57] |

[a] All is referred to data from both periods together: 19 June 2015 to 31 October 2015 and from 1 April 2016 to 31 October 2016.





**Table 7. Comparison of the performance of modelled interception loss using different parametrization. Data-set 1 refers to the period from 19 June 2015 to 31 October 2015, and data-set 2 to the period from 1 April 2016 to 31 October 2016.**

| Data | Description | Parametrization | | | | | Relative error (%) | RMSE | Nash-Sutcliffe |
|---|---|---|---|---|---|---|---|---|---|
| | | $S$ | $p$ | $\overline{E}/\overline{R}$ | $I$ (%) | $I$ (mm) | | | |
| Data-set 1 (Calibration) | Run 1 | 1.37 | 0.28 | 0.23 | 36.3 | 175.5 | -2.75 | 0.93 | 0.90 |
| | Run 2 | 1.90 | 0.17 | 0.19 | 34.4 | 166.6 | -7.70 | 1.36 | 0.79 |
| | Measured $I$ | | | | 37.3 | 180.4 | | | |
| Data-set 2 (Validation) | | | | | | | | | |
| | Run 1V | 1.37 | 0.28 | 0.23 | 40.43 | 108.66 | -0.77 | 0.78 | 0.79 |
| | Run 2V | 1.90 | 0.17 | 0.15 | 40.31 | 108.35 | -1.06 | 0.79 | 0.79 |
| | Measured $I$ | | | | 40.74 | 109.5 | | | |



**Table 8. Components of interception loss in mm (and as percentage of total) for data-set 2 (19 June 206 to 31 October 2016) based on the validated Gash analytical original model.**

| Interception component mm (%) | Run 1 | Run 2 |
|---|---|---|
| $m$ storms with $P_G$ insufficient to saturate the canopy ($P_G < P_G'$) | 18.9 mm (17.5 %) | 25.9 mm (23.9 %) |
| Wetting up the canopy with n storms large enough to saturate the canopy ($P_G \geq P_G'$) | 7.9 mm (7.4 %) | 5.1 mm (4.7 %) |
| Evaporation from the saturated canopy during rainfall | 40.53 mm (37.3 %) | 25.6 mm (23.6 %) |
| Evaporation after rainfall event | 36.9 mm (34.1 %) | 47.5 mm (43.8 %) |
| Evaporation from trunks, saturated and non-saturated conditions | 4.2 mm (3.8 %) | 4.2 mm (3.8 %) |



**Table 9. Summary of canopy properties and interception parameters for Douglas-fir forests.**

| Reference | Age (year) | Height (m) | Density (tree ha⁻¹) | LAI (m m⁻¹) | $I$ (%) | $S$ (mm) | $E$ (mm h⁻¹) | Reference |
|---|---|---|---|---|---|---|---|---|
| US (NW Pacific)[a] | 25 | 20 | 2200 | 10 | 21 | 1.3 | 0.25 | Pypker et al. (2005) |
| Netherlands | 29 | 18 | 992 | 8[b] | 38 | 2.4 | 0.077 | Klaassen et al. (1998) |
| UK | 42 | 24 | 660 | 12 | 39 | 2.1 | NA | Rutter et al. (1975) |
| Netherlands | 55 | 34 | 570 | 4.5 | 34 | 1.7 | 0.20 | This Study |
| Belgium | 80 | 41 | 145 | 4.2 | 30 | NA | NA | Soubie et al. (2016) |
| US (NW Pacific)[a] | >450 | 60 | 560 | 8.6 | 24 | 2.7-4.2 | 0.14 | Link et al. (2004) |
| US (NW Pacific)[a] | >450 | 39 | 441 | 9.6 | 24 | 3.32 | 0.21 | Pypker et al. (2005) |

[a] Mixed Douglas-fir and Western hemlock

[b] Klaasen et al. (1998) reported an LAI measured by destructive method, but LAI estimated with Li-Cor2000 was 8 (cf. Heij and Schneider, 1991)



**Table A1. Statistical description of collection periods of throughfall and average amounts for a sample size $n$=32.**

| Per. | Date | Sampling distribution | Days in period | Cumulative average $Tf$ Funnels (mm) ($\pm$SD) | | Cumulative $Tf$ gutters (mm) |
|---|---|---|---|---|---|---|
| 1 | 17-Feb to 17-Mar-2015 | Roving | 28 | 36.6 | ( 6.6) | 33.6 |
| 2 | 17-Mar to 03-Apr-2015 | Roving | 17 | 43.7 | (5.6) | 42.9 |
| 3 | 03-Apr to 01-May-2015 | Roving | 28 | 7.3 | (1.9) | 8.7 |
| 4 | 01-May to 16-May-2015 | Roving | 15 | 8.7 | (1.5) | 8.2 |
| 5 | 16-May to 30-May-2015 | Roving | 14 | 6.7 | (2.0) | 6.6 |
| 6 | 30-May to 12-Jun-2015 | Roving | 13 | 11.2 | (1.7) | 11.0 |
| 7 | 12-Jun to 29-Jun-2015 | Roving | 17 | 37.3 | (6.4) | 38.3 |
| 8 | 29-Jun to 15-Jul-2015 | Roving | 16 | 20.0 | (3.7) | 20.1 |
| 9 | 15-Jul to 01-Aug-2015 | Roving | 17 | 74.1 | (8.0) | 70.2 |
| 10 | 01-Aug to 15-Aug-2015 | Roving | 14 | 7.4 | (1.8) | 7.3 |
| 11 | 15-Aug to 28-Aug-2015 | Non-roving | 13 | 71.4 | (8.7) | 69.6 |
| 12 | 28-Aug to 15-Sep-2015 | Non-roving | 18 | 62.4 | (7.0) | 60.4 |
| 13 | 15-Sep to 29-Sep-2015 | Non-roving | 14 | 16.2 | (2.2) | 15.3 |
| 14 | 29-Sep to 19-Oct-2015 | Non-roving | 20 | 25.1 | (2.8) | 24.4 |
| 15 | 19-Oct to 02-Nov-2015 | Non-roving | 14 | 2.6 | (0.8) | 2.0 |