# Peer review of "The influence of long-term changes in canopy structure on rainfall interception loss: a case study in Speulderbos, the Netherlands."

_Hydrology and Earth System Sciences, 2018_

## Referee Comment (RC1) · Anonymous Referee #1 · 2 Apr 2018

This study provide a comprehensive measurements and modelling of forest water and energy budgets, and discussed in depth on the canopy interception of rainfall and subsequent evaporation from interception, in comparison to estimates in the same but younger forest stand. Canopy interception is of course an important component in vegetated surface water balance as it can account for a large proportion of gross precipitation, and thus affect soil and groundwater recharge, storage, and catchment discharge. One of the difficulties in studying interception is that it is subject to many interactive factors including climatic factors as mentioned in this paper the wind and rainfall intensity etc., and forest structures and species composition.

[Figure]

Regarding this paper, it is well written, methods used are appropriate, data analyses and interpretations sufficiently accommodate the results and discussion, in particular the possible reasons why evaporation was lower when canopy was denser and canopy storage capacity was higher, which is important to clarify. I have no major comments therefore but only a few minor for the authors to consider corrections, given below.

1. Page 9, Line 11. You talked 'the performance of the sonic anemometer (CSAT3) during wet conditions was evaluated by...' Can you given the reason why doing this right after this sentence, maybe in just one sentence.

2. Page 11, Line 20. You referenced Fig 4c, but there is no such figure. Please provide it. Line 23, IET method gave mean values of p in Table 4 is 0.17, but in text is 0.22. Double check.

3. Page 12, use of figure numbers: you may want to swap Fig No. 7 and 8 as the latter appears first in the text.

4. Page 2, Line 24: insert 'such' before 'as'

5. Page 3, Line 26: as DBH appears for the first time in the text, expand in full here rather than Page 5 Line 7-8.

6. Page 4, Line 23: insert 'm' after 32 and before 'x 64 m'

7. Page 13, Line 14: delete 'predicted'

8. Page 5, Line 9: insert 'mm' for 0.14

Figures: please increase marker size in Fig 4, 5 for the dots for better visual.

Table 9: unit of LAI: m2 m-2

---

## Referee Comment (RC2) · Anonymous Referee #2 · 22 Apr 2018

This manuscript reports on wet-canopy evaporation research conducted in a well-instrumented site that has also been the site of important work on the same topic in previous decades. In general the data and analyses appear to be of high quality. The manuscript is rich in detail, but in places it focuses on presenting details at the expense of a comprehensive logically coherent examination of the objectives.

The objectives themselves would benefit from some clarification. I think objective ii could be phrased better, in that the source of the latent heat flux is known but the real question is the source of available energy to drive that latent flux. Objectives iii and iv are too general to be useful. Aligning objectives with motivating statements

would also improve clarity. For example, P2L31 suggests the objectives will include analysis of how canopy variation affects wet-canopy evaporation over time, and indeed there are some such comparisons in the discussion, but there are no explicit objectives pertaining to this goal, and only one very general conclusion about it.

One aspect of the work that I think warrants more robust treatment is the role of the canopy in supplying available energy for evaporation. There is little discussion of the sensitivity of various assumptions needed to support the estimations of this energy, yet it ends up being a relatively large proportion of the total budget.

There is quite a bit of duplicative text. For example, the insensitivity of interception models to in-storm vs. post-storm evaporation (i.e., Fig. 10) is mentioned at least four separate times in the introduction, results, and discussion. I think perhaps 3-5 tables and figures could be eliminated to reduce the overwhelming detail (some candidates: F2, F3, F9, combine F4 and F5; T3, T4 to text).

P6L13-14 both citations of tree properties should be to the primary sources rather than these secondary works.

P7 section 2.3 needs a name like "modeling," or perhaps it should not exist and section 2.3.1 should be 2.2.8.

P7L25 does this mean data outside these two windows were completely omitted from all analyses? It doesn't seem so.

P8L3 and P9L6 the meaning of "was preferred" is not clear in either place; please state exactly what was done instead of what was desired. Does this mean that Ebar is not average as stated P9L4?

P11L3 this information duplicates the methods.

P11L5 how can SE be calculated from accuracy?

P11L29 it is not clear what the denominator is in this percentage

[Figure]

P12L12 is this not the "'water balance based' evaporation rate" promised P9L7?

P13L30-31 redundant with figure caption

P16L30 the splash droplet hypothesis does not depend on high rainfall intensities. Its mention here is also unrelated to the rest of the paragraph.

Table 2 first column heading is mistakenly labeled "data-logger"

Table 7, Fig 10 the meaning of Run 1 and Run 2 must be specified here

---

## Author Comment (AC1) · 31 May 2018

Reply to Referee 1

This study provide a comprehensive measurements and modelling of forest water and energy budgets, and discussed in depth on the canopy interception of rainfall and subsequent evaporation from interception, in comparison to estimates in the same but younger forest stand. Canopy interception is of course an important component in vegetated surface water balance as it can account for a large proportion of gross precipitation, and thus affect soil and groundwater recharge, storage, and catchment discharge. One of the difficulties in studying interception is that it is subject to many interactive fac-

tors including climatic factors as mentioned in this paper the wind and rainfall intensity etc., and forest structures and species composition. Regarding this paper, it is well written, methods used are appropriate, data analyses and interpretations sufficiently accommodate the results and discussion, in particular the possible reasons why evaporation was lower when canopy was denser and canopy storage capacity was higher, which is important to clarify. I have no major comments therefore but only a few minor for the authors to consider corrections, given below

We would like to thank Reviewer1 for being interested and recognizing the value of our paper. In the following, we respond to each individual comment.

1. Page 9, Line 11. You talked 'the performance of the sonic anemometer (CSAT3) during wet conditions was evaluated by. . .' Can you given the reason why doing this right after this sentence, maybe in just one sentence.

This is a good suggestion and we thank Reviewer1 for this. We have added the following sentence in the revised version:"According to Monin-Obukhov similarity theory $\sigma w/u^*$ in neutral conditions is a universal constant, therefore the ability of the anemometer to measure $\sigma w/u^*$ during wet and dry conditions was tested (Gash et al., 1999)".

2. Page 11, Line 20. You referenced Fig 4c, but there is no such figure. Please provide it. Line 23, IET method gave mean values of p in Table 4 is 0.17, but in text is 0.22. Double check.

In response to the first part of the comment: thank you for pointing this out. Following the comments and suggestions from Reviewer2, we have decided to reorganize the figures and remove Figure 4c from the manuscript. We also removed the reference to this figure. In response to the second part of the comment: this was a mistake. The correct value for p using IEA method is 0.17. We have corrected this in the revised version.

3. Page 12, use of figure numbers: you may want to swap Fig No. 7 and 8 as the latter
appears first in the text.

Thank you for pointing this out - we have corrected the sequence of the figures in the revised manuscript.

4. Page 2, Line 24: insert 'such' before 'as'

Done.

5. Page 3, Line 26: as DBH appears for the first time in the text, expand in full here rather than Page 5 Line 7-8

Done

6. Page 4, Line 23: insert 'm' after 32 and before 'x 64 m'

Done.

7. Page 13, Line 14: delete 'predicted'

Done.

8. Page 5, Line 9: insert 'mm' for 0.14

Done (in page 15, Line 9)

Figures: please increase marker size in Fig 4, 5 for the dots for better visual. Table 9: unit of LAI: m2 m-2

Done.

References

Gash, J., Valente, F., and David, J. S.: Estimates and measurements of evaporation from wet, sparse pine forest in Portugal, Agricultural and Forest Meteorology, 94, 149-158, http://dx.doi.org/10.1016/S0168-1923(99)00008-8, 1999.

54, 2018.

---

## Author Comment (AC2) · 31 May 2018

This manuscript reports on wet-canopy evaporation research conducted in a well instrumented site that has also been the site of important work on the same topic in previous decades. In general the data and analyses appear to be of high quality. The manuscript is rich in detail, but in places it focuses on presenting details at the expense of a comprehensive logically coherent examination of the objectives.

We sincerely thank Reviewer#2 for the detailed comments and valuable suggestions which helped us very much to improve the manuscript. In the following, we respond to the comments one by one.

[Figure]

The objectives themselves would benefit from some clarification. I think objective ii could be phrased better, in that the source of the latent heat flux is known but the real question is the source of available energy to drive that latent flux. Objectives iii and iv are too general to be useful. Aligning objectives with motivating statements would also improve clarity. For example, P2L31 suggests the objectives will include analysis of how canopy variation affects wet-canopy evaporation over time, and indeed there are some such comparisons in the discussion, but there are no explicit objectives pertaining to this goal, and only one very general conclusion about it.

This is a very good suggestion indeed, and we thank Reviewer#2 for this. We have rephrased the objective (ii). Moreover, the objectives (iii) and (iv) have been revised to make them more clear and specific. The revised objectives are now properly aligned with the motivating statement in the introduction section. The revised objectives read:

i) assess two indirect methods for estimating canopy storage capacity;

ii) quantify the sources of energy that drive the latent heat flux involved in the evaporation of intercepted rainfall;

iii) examine the effect of long-term changes in canopy structure on the rainfall interception losses;

iv) explore the relative importance of climatic and forest structural factors to overall rainfall interception loss using a physically based interception model.

One aspect of the work that I think warrants more robust treatment is the role of the canopy in supplying available energy for evaporation. There is little discussion of the sensitivity of various assumptions needed to support the estimations of this energy, yet it ends up being a relatively large proportion of the total budget.

Indeed, one of the important sources of energy for evaporation of intercepted rainfall is the energy stored in the canopy. Although it is not the largest one (only represents 15 % of the total energy involved), in the revised version, we have now discussed the

sensitivity of the canopy biomass released energy to the assumptions considered in our calculations.

We have done an error propagation exercise, applied to Eq. 3 which embrace most of our assumptions. The uncertainty will be the quadratic sum of the relative errors $\delta m_{bio}$, $\delta c_v$, and $\delta \Delta T_{bio}$. The $\delta m_{bio}$ is related mainly to the uncertainty of allometric equations (18% for n = 23, Chave et al., 2004) in combination with the uncertainty in the assumed moisture content (ranging from 44 % to 55 %). Then the combined uncertainty for $\delta m_{bio}$ would be about 27 %. Regarding $c_v$, the range of values used in studies with similar species is from 2400 to 2928 (J kg-1 K-1) (Oliphant et al., 2004), which means an uncertainty of 22%. The uncertainty of $\Delta T_{bio}$ assumed to be equal to $\Delta T_{air}$ would be the largest one. Based on data presented by Meesters and Vugts (1996) (their Fig.6 ) the difference in temperature amplitude between $T_{air}$ and $T_{bio}$ would yield to an uncertainty of about 40 %. Then the error propagation of the product of the three variables will yield a 53 % of uncertainty for $Q_{bio}$.

There is quite a bit of duplicative text. For example, the insensitivity of interception models to in-storm vs. post-storm evaporation (i.e., Fig. 10) is mentioned at least four separate times in the introduction, results, and discussion. I think perhaps 3-5 tables and figures could be eliminated to reduce the overwhelming detail (some candidates: F2, F3, F9, combine F4 and F5; T3, T4 to text).

We have improved the readability of the manuscript by taking out any duplication and reducing the number of figures and tables.

Following the comments and suggestions from Reviewer 2, we have removed Figures 2, 3 and 9. Similarly, we have combined Figures 4 and 5 into a single figure (Figure 2 in the revised version). As for the tables, we have removed the Table 4 but have decided to keep the Table 3 for readability as it contains the main equations used in the Gash model to quantify the different components of interception loss.

P6L13-14 both citations of tree properties should be to the primary sources rather than

these secondary works.

Thank you for pointing this out, in the revised version we have corrected the citation for specific heat of biomass ($c_v$), the primary source is indeed Michiles and Gielow (2008). And for the value of moisture content, we have verified and properly cited the work of Nord-Larsen and Nielsen (2015) that is the primary source.

P7 section 2.3 needs a name like "modeling," or perhaps it should not exist and section 2.3.1 should be 2.2.8.

We agree and have renamed the section headings: 2.3 Modelling rainfall interception and 2.3.1 The Gash rainfall interception model

P7L25 does this mean data outside these two windows were completely omitted from all analyses? It doesn't seem so.

Data outside those two windows were not omitted from all analysis. They were omitted only for the modelling purposes. We have now devoted some texts in the modelling section to clarify this issue.

P8L3 and P9L6 the meaning of "was preferred" is not clear in either place; please state exactly what was done instead of what was desired. Does this mean that Ebar is not average as stated P9L4?

Thank you for pointing this out. We have removed that sentence from P8L3 to avoid duplicative text. In P9L6, We have revised the paragraph. It now reads: "Because the distribution of rainfall intensity was highly skewed, we used the median rainfall intensity following the recommendations of Schellekens et al. (1999). The thus derived value of $\bar{E}$ will henceforth be referred to as the 'water balance based' evaporation rate." As for the last part of the comment, we have elaborated the waterbalance approach (P9L4) to make this clear.

P11L3 this information duplicates the methods.

We have removed the sentence in the revised version.

P11L5 how can SE be calculated from accuracy?

This was a mistake, and we thank Reviewer#2 for pointing this out. In the revised version we have calculated the SE for our Sf measurements correctly and have revised the sentence accordingly.

P11L29 it is not clear what the denominator is in this percentage

The denominator is Rn-G-Q, the available energy for the turbulent heat fluxes H +LE. In fact, the percentage was calculated as 100% *slope, where the slope was derived from the regression of H+LE versus Rn-G-Q. In the revised version, we will keep only the slope value as an indicator of the energy balance closure to avoid any misunderstanding.

P12L12 is this not the "'water balance based' evaporation rate" promised P9L7?

Yes, that is water balance based evaporation rate as mentioned in P12L9. We have revised the sentence for greater clarity. It now reads: "The parameter $\bar{E}/\bar{R}$ , multiplied by the median R of 0.82 mm h-1, results in an estimated water-balance based evaporation rate of 0.19 mm h-1."

P13L30-31 redundant with figure caption

We have revised the sentence. It now reads: "The sensitivity analysis of the Gash model shows that parameter equifinality (Beven, 1993) occurs between S and $\bar{E}/\bar{R}$ (van Dijk et al., 2015), which implies in this case that an underestimation of S is likely to be compensated by overestimation of $\bar{E}/\bar{R}$

P16L30 the splash droplet hypothesis does not depend on high rainfall intensities. Its mention here is also unrelated to the rest of the paragraph.

We apologise for not having been clear enough in the original version of our manuscript. What we wanted to say in P6L30 was enhanced evaporation of rain

droplets splashed from the tree canopy. And it is well known that the specific number and the average size of raindrops increase with rainfall intensity (Murakami, 2006). We have revised the sentence, it now reads: "Likewise, enhanced evaporation of rain droplets splashed from the tree canopy has been invoked as a possible mechanism allowing high interception losses (Murakami, 2006) but given the low rainfall intensities prevailing in the study area this is not likely to be important."

Table 2 first column heading is mistakenly labelled "data-logger"

We have corrected this in the revised version.

Table 7, Fig 10 the meaning of Run 1 and Run 2 must be specified here

We have described Run 1 and Run 2 in the respective figure/table captions of the revised manuscript.

References

Beven, K.: Prophecy, reality and uncertainty in distributed hydrological modelling, Advances in Water Resources, 16, 41-51, https://doi.org/10.1016/0309-1708(93)90028-E, 1993.

Chave, J., Condit, R., Aguilar, S., Hernandez, A., Lao, S., and Perez, R.: Error propagation and scaling for tropical forest biomass estimates, Philosophical Transactions of the Royal Society B: Biological Sciences, 359, 409-420, 2004. Meesters, A. G. C. A., and Vugts, H. F.: Calculation of heat storage in stems, Agricultural and Forest Meteorology, 78, 181-202, http://dx.doi.org/10.1016/0168-1923(95)02251-1, 1996.

Michiles, A. A. d. S., and Gielow, R.: Above-ground thermal energy storage rates, trunk heat fluxes and surface energy balance in a central Amazonian rainforest, Agricultural and Forest Meteorology, 148, 917-930, http://dx.doi.org/10.1016/j.agrformet.2008.01.001, 2008.

Murakami, S.: A proposal for a new forest canopy interception mechanism: Splash droplet evaporation, Journal of Hydrology, 319, 72-82, http://dx.doi.org/10.1016/j.jhydrol.2005.07.002, 2006.

Nord-Larsen, T., and Nielsen, A. T.: Biomass, stem basic density and expansion factor functions for five exotic conifers grown in Denmark, Scandinavian Journal of Forest Research, 30, 135-153, http://dx.doi.org/10.1080/02827581.2014.986519, 2015.

Oliphant, A. J., Grimmond, C. S. B., Zutter, H. N., Schmid, H. P., Su, H. B., Scott, S. L., Offerle, B., Randolph, J. C., and Ehman, J.: Heat storage and energy balance fluxes for a temperate deciduous forest, Agricultural and Forest Meteorology, 126, 185-201, http://dx.doi.org/10.1016/j.agrformet.2004.07.003, 2004.

van Dijk, A. I. J. M., Gash, J. H., van Gorsel, E., Blanken, P. D., Cescatti, A., Emmel, C., Gielen, B., Harman, I. N., Kiely, G., Merbold, L., Montagnani, L., Moors, E., Sottocornola, M., Varlagin, A., Williams, C. A., and Wohlfahrt, G.: Rainfall interception and the coupled surface water and energy balance, Agricultural and Forest Meteorology, 214–215, 402-415, http://dx.doi.org/10.1016/j.agrformet.2015.09.006, 2015.
* * *

---

## Editor Comment (EC1) · N. Ursino (Editor) · 12 Jun 2018

I would kindly solicitate the Reviewers and in particular Anonymous Referee #2, to express in form of open discussion their appreciation for the suggested changes reported by Vaca et al. in their replyto Reviewers' comments. Did the Authors address all your concerns? Nadia Ursino

---

## Author Response (AR1)

**Response letter – manuscript HESS-2018-54**

The comments provided by Editor and Referees have been reported below as italicized text. Our response follows point-by-point.

**EDITOR COMMENT**

> *Dear Authors,*
> *Your reply convinced me and the referees.*
> *Please, address all referees' comments according to your reply in a new version of your manuscript and proceed with the upload.*
> *Sincerely,*
> *Nadia Ursino*

We acknowledge Dr. Ursino for managing the review process. We truly appreciated the comments provided by the referees. The manuscript has been modified to accommodate the referees' s minor suggestions.

**Reply to Referee #1**

> *This study provides comprehensive measurement and modelling of forest water and energy budgets, and discussed in depth on the canopy interception of rainfall and subsequent evaporation from interception, in comparison to estimates in the same but younger forest stand. Canopy interception is of course an important component in vegetated surface water balance as it can account for a large proportion of gross precipitation, and thus affect soil and groundwater recharge, storage, and catchment discharge. One of the difficulties in studying interception is that it is subject to many interactive factors including climatic factors as mentioned in this paper the wind and rainfall intensity etc., and forest structures and species composition.*
>
> *Regarding this paper, it is well written, methods used are appropriate, data analyses and interpretations sufficiently accommodate the results and discussion, in particular the possible reasons why evaporation was lower when canopy was denser and canopy storage capacity was higher, which is important to clarify. I have no major comments therefore but only a few minor for the authors to consider corrections, given below*

We thank Referee#1 for being interested and recognizing the value of our paper. In the following, we respond to each comment.

> *1. Page 9, Line 11. You talked 'the performance of the sonic anemometer (CSAT3) during wet conditions was evaluated by. . .' Can you gave the reason why doing this right after this sentence, maybe in just one sentence.*

This is a good suggestion. We have added the following sentence in the revised version: "According to Monin-Obukhov similarity theory $\sigma w/u^*$ in neutral conditions is a universal constant, therefore the ability of the anemometer to measure $\sigma w/u^*$ during wet and dry conditions was tested (Gash et al., 1999)".

> *2. Page 11, Line 20. You referenced Fig 4c, but there is no such figure. Please provide it. Line 23, IET method gave mean values of p in Table 4 is 0.17, but in text is 0.22. Double check.*

In response to the first part of the comment: Following the comments and suggestions from Referee#2, we have decided to reorganize the figures and remove Figure 4c from the manuscript. We also removed the reference to this figure.
In response to the second part of the comment: this was a mistake. The correct value for *p* using IEA method is 0.17. We have corrected this in the revised version.

*3. Page 12, use of figure numbers: you may want to swap Fig No. 7 and 8 as the latter appears first in the text.*

We have corrected the sequence of the figures in the revised manuscript.

*4. Page 2, Line 24: insert 'such' before 'as'*

Done.

*5. Page 3, Line 26: as DBH appears for the first time in the text, expand in full here rather than Page 5 Line 7-8*

Done

*6. Page 4, Line 23: insert 'm' after 32 and before 'x 64 m'*

Done.

*7. Page 13, Line 14: delete 'predicted'*

Done.

*8. Page 5, Line 9: insert 'mm' for 0.14*

Done (in page 15, Line 9)

*Figures: please increase marker size in Fig 4, 5 for the dots for better visual. Table 9: unit of LAI: m2 m-2*

Done.

**Reply to Referee #2**

*This manuscript reports on wet-canopy evaporation research conducted in a well-instrumented site that has also been the site of important work on the same topic in previous decades. In general the data and analyses appear to be of high quality. The manuscript is rich in detail, but in places it focuses on presenting details at the expense of a comprehensive logically coherent examination of the objectives.*

We sincerely thank Referee #2 for the detailed comments and valuable suggestions which helped us very much to improve the manuscript. In the following, we respond to the comments one by one.

*The objectives themselves would benefit from some clarification. I think objective ii could be phrased better, in that the source of the latent heat flux is known but the real question is the source of available energy to drive that latent flux. Objectives iii and iv are too general to be useful. Aligning objectives with motivating statements would also improve clarity. For example, P2L31 suggests the objectives will include analysis of how canopy variation affects wet-canopy evaporation over time, and indeed there are some such comparisons in the discussion, but there are no explicit objectives pertaining to this goal, and only one very general conclusion about it.*

This is a very good suggestion indeed. We have rephrased the objective (ii). Moreover, the objectives (iii) and (iv) have been revised to make them more clear and specific. The revised objectives are now properly aligned with the motivating statement in the introduction section. The revised objectives read:

i) assess two indirect methods for estimating canopy storage capacity;

ii) quantify the sources of energy that drive the latent heat flux involved in the evaporation of intercepted rainfall;

iii) examine the effect of long-term changes in canopy structure on the rainfall interception losses;

iv) explore the relative importance of climatic and forest structural factors to overall rainfall interception loss using a physically based interception model.

*One aspect of the work that I think warrants more robust treatment is the role of the canopy in supplying available energy for evaporation. There is little discussion of the sensitivity of various assumptions needed to support the estimations of this energy, yet it ends up being a relatively large proportion of the total budget.*

Indeed, one of the important sources of energy for evaporation of intercepted rainfall is the energy stored in the canopy. In the revised version, we have now quantified and discussed the sensitivity of the canopy biomass released energy to the assumptions considered in our calculations.

We carried out an error propagation exercise, applied to Eq. 3 which embrace most of our assumptions. The uncertainty will be the quadratic sum of the relative errors $\delta m_{bio}$, $\delta c_v$, and $\delta\Delta T_{bio}$. The $\delta m_{bio}$ is related mainly to the uncertainty of allometric equations (18% for n = 23, Chave et al., 2004) in combination with the uncertainty in the assumed moisture content (ranging from 44 % to 55 %). Then the combined uncertainty for $\delta m_{bio}$ would be about 27 %. Regarding $c_v$, the range of values used in studies with similar species is from 2400 to 2928 (J kg$^{-1}$ K$^{-1}$) (Oliphant et al., 2004), which means an uncertainty of 22%. $\Delta T_{bio}$, which is assumed to be equal to $\Delta T_{air}$, has the largest uncertainty.  Based on data presented by Meesters and Vugts (1996) (their Fig.6 ) the difference in temperature amplitude between $T_{air}$ and $T_{bio}$ would yield to an uncertainty of about 40 %. The error propagation of the product of the three variables yields an uncertainty for $Q_{bio}$ of 53%.

> *There is quite a bit of duplicative text. For example, the insensitivity of interception models to in-storm vs. post-storm evaporation (i.e., Fig. 10) is mentioned at least four separate times in the introduction, results, and discussion. I think perhaps 3-5 tables and figures could be eliminated to reduce the overwhelming detail (some candidates: F2, F3, F9, combine F4 and F5; T3, T4 to text).*

We have improved the readability of the manuscript by taking out any duplication and reducing the number of figures and tables. Following the comments and suggestions from Referee #2, we have removed Figures 2, 3 and 9. Similarly, we have combined Figures 4 and 5 into a single figure (Figure 2 in the revised version). As for the tables, we have removed the Table 4 but have decided to keep the Table 3 for readability as it contains the main equations used in the Gash model to quantify the different components of interception loss.

> *P6L13-14 both citations of tree properties should be to the primary sources rather than these secondary works.*

Thank you for pointing this out, in the revised version we have corrected the citation for the specific heat of biomass ($c_v$), the primary source is indeed Michiles and Gielow (2008). And for the value of moisture content, we have verified and properly cited the work of Nord-Larsen and Nielsen (2015) that is the primary source.

> *P7 section 2.3 needs a name like "modeling," or perhaps it should not exist and section 2.3.1 should be 2.2.8.*

We agree and have renamed the section headings: 2.3 Modelling rainfall interception and 2.3.1 The Gash rainfall interception model

> *P7L25 does this mean data outside these two windows were completely omitted from all analyses? It doesn't seem so.*

Data outside those two windows were not omitted from all analysis. They were omitted only for the modelling purposes. We have now devoted some text in the modelling section to clarify this issue.

> *P8L3 and P9L6 the meaning of "was preferred" is not clear in either place; please state exactly what was done instead of what was desired. Does this mean that Ebar is not average as stated P9L4?*

Thank you for pointing this out. To avoid duplicative text, we have removed that sentence from P8L3. In P9L6, We have revised the paragraph. It now reads: "Because the distribution of rainfall intensity was highly skewed, we used the median rainfall intensity following the recommendations of Schellekens et al. (1999). The thus derived value of $\bar{E}$ will henceforth be referred to as the 'water balance based' evaporation rate."
As for the last part of the comment, we have elaborated the water-balance approach (P9L4) to make this clear.

> *P11L3 this information duplicates the methods.*

We have removed the sentence in the revised version.

*P11L5 how can SE be calculated from accuracy?*

This was a mistake, and we thank Referee #2 for pointing this out. In the revised version we have calculated the SE for our Sf measurements correctly and have revised the sentence accordingly.

*P11L29 it is not clear what the denominator is in this percentage*

The denominator is Rn-G-Q, the available energy for the turbulent heat fluxes H +LE. In fact, the percentage was calculated as 100% *slope, where the slope was derived from the regression of H+LE versus Rn-G-Q. In the revised version, we will keep only the slope value as an indicator of the energy balance closure to avoid any misunderstanding.

*P12L12 is this not the "'water balance based' evaporation rate" promised P9L7?*

Yes, that is water balance based evaporation rate as mentioned in P12L9. We have revised the sentence for clarification. It now reads: "The parameter $\bar{E}/\bar{R}$, multiplied by the median $R$ of 0.82 mm h$^{-1}$, results in an estimated water-balance based evaporation rate of 0.19 mm h$^{-1}$."

*P13L30-31 redundant with figure caption*

We have revised the sentence. It now reads: "The sensitivity analysis of the Gash model shows that parameter equifinality (Beven, 1993) occurs between $S$ and $\bar{E}/\bar{R}$ (van Dijk et al., 2015), which implies in this case that an underestimation of $S$ is likely to be compensated by overestimation of $\bar{E}/\bar{R}$ "

*P16L30 the splash droplet hypothesis does not depend on high rainfall intensities. Its mention here is also unrelated to the rest of the paragraph.*

We apologise for not having been clear enough in the original version of our manuscript. What we wanted to say in P6L30 was that rain droplets splashed from the tree canopy may enhance evaporation. And it is well known that the specific number and the average size of raindrops increase with rainfall intensity (Murakami, 2006). We have revised the sentence, it now reads: "Likewise, enhanced evaporation of rain droplets splashed from the tree canopy has been mentioned as a possible mechanism allowing high interception losses (Murakami, 2006) but given the low rainfall intensities prevailing in the study area this is not likely to be important."

*Table 2 first column heading is mistakenly labelled "data-logger"*

We have corrected this in the revised version.

*Table 7, Fig 10 the meaning of Run 1 and Run 2 must be specified here*

We have described Run 1 and Run 2 in the respective figure/table captions of the revised manuscript.

**References**

[revised manuscript text omitted]